# Repair of airway epithelia requires metabolic rewiring towards fatty acid oxidation

Stefania Crotta [1,7] ✉, Matteo Villa [2,7], Jack Major[1], Katja Finsterbusch [1], Miriam Llorian[3], Peter Carmeliet [4,5,6], Joerg Buescher [2] & Andreas Wack [1] ✉

Epithelial tissues provide front-line barriers shielding the organism from invading pathogens and harmful substances. In the airway epithelium, the combined action of multiciliated and secretory cells sustains the mucociliary escalator required for clearance of microbes and particles from the airways. Defects in components of mucociliary clearance or barrier integrity are associated with recurring infections and chronic inflammation. The timely and balanced differentiation of basal cells into mature epithelial cell subsets is therefore tightly controlled. While different growth factors regulating progenitor cell proliferation have been described, little is known about the role of metabolism in these regenerative processes. Here we show that basal cell differentiation correlates with a shift in cellular metabolism from glycolysis to fatty acid oxidation (FAO). We demonstrate both in vitro and in vivo that pharmacological and genetic impairment of FAO blocks the development of fully differentiated airway epithelial cells, compromising the repair of airway epithelia. Mechanistically, FAO links to the hexosamine biosynthesis pathway to support protein glycosylation in airway epithelial cells. Our findings unveil the metabolic network underpinning the differentiation of airway epithelia and identify novel targets for intervention to promote lung repair.

The lung is a vital organ whose gas-exchange function must be maintained for efficient oxygen delivery to distal tissue sites, as damage to organs such as brain and kidney sets in after only minutes of insufficient oxygen supply[1]. Efficient repair processes must therefore be in place to ensure functionality after damage caused by infections, chemicals, or other injury. The lung is also a unique environment from a metabolic point of view, with a degree of oxygenation that far exceeds that in other organs[2]. While much is known about lung development, the mechanisms and metabolic changes underpinning the repair of adult lung tissue upon injury are less clear[3–5]. As respiratory diseases represent a leading cause of death worldwide[6], a better knowledge of repair processes in this unique metabolic environment is urgently needed.

Compared to the intestine and the haematopoietic system, lungs have a complex stem/progenitor cell hierarchy and at steady state exhibit a low cell turnover; yet, after injury, depending on the severity of the insult, different progenitor cells are activated in the lung to respond rapidly and robustly and replace damaged cells[4,5,7]. This in combination with the phenomena of transdifferentiation and de-differentiation often renders the interpretation of lung epithelial repair processes difficult[5]. Airway epithelia are pseudostratified and mainly consist of p63-positive basal cells (BCs), a variety of secretory cells including MucA/C-producing goblet cells responsible for mucus generation, CCSP-expressing Club cells, and Foxj1-positive multiciliated cells whose co-ordinated ciliary movement sets in motion the

[1]Immunoregulation Laboratory, Francis Crick Institute, London, UK. [2]Max Planck Institute of Immunobiology and Epigenetics, Freiburg, Germany. [3]Bioinformatics, Francis Crick Institute, London, UK. [4]Laboratory of Angiogenesis and Vascular Metabolism, Center for Cancer Biology, VIB, and Department of Oncology, KU Leuven, Leuven, Belgium. [5]Laboratory of Angiogenesis & Vascular Heterogeneity, Department of Biomedicine, Aarhus University, Aarhus, Denmark. [6]Center for Biotechnology (BTC), Khalifa University of Science and Technology, PO Box 127788 Abu Dhabi, United Arab Emirates. [7]These authors contributed equally: Stefania Crotta, Matteo Villa. ✉e-mail: Stefania.Crotta@crick.ac.uk; Andreas.Wack@crick.ac.uk

mucociliary escalator transporting mucus to the oesophagus where it is cleared.

Basal cells constitute an extended population of multipotent stem cells that regulate both homeostasis of the tissue and its regeneration after injury[8]. Repair of damaged airway epithelia is a complex process that requires a tight balance between BC self-renewal and generation of physiologically appropriate numbers of differentiated secretory and multiciliated cells. This process is highly regulated as imperfect outcome could lead to the activation of fibrotic or metaplastic responses[8,9].

Upon injury at barrier sites, stromal, parenchymal and immune cells undergo substantial metabolic changes to support repair and recovery[10–12]. Immune cells change metabolism while moving between resting and activated states or from inflammatory towards pro-resolution phenotypes[13,14]. Nutrient availability and immune signalling both regulate such metabolic changes that, in turn, are key determinants of immune cell functions[10]. Similarly, changes in endothelial metabolism have been linked to vascular protection and angiogenesis[15–17].

So far, a reliance on glucose and glutamine metabolism has been shown for alveolar regeneration in alveolar type II (AT2) cells[18,19]. In addition, the regenerative capacity of a specific subset of airway progenitor cells identified as variant Club cells has been shown to depend on autophagy, through regulation of glucose metabolism[20]. However, it is unclear whether changes in cellular metabolism play a role in coordinating the development or the post-injury repair of airway epithelia.

Cellular metabolism is a complex network that adapts to local substrate availability to fulfil cellular requirements for energy, redox balance and biomass generation. Glucose is generally regarded as a central energy source for cells as it fuels glycolysis, generating pyruvate that provides carbons for the tricarboxylic acid (TCA) cycle which, in turn, yields reducing equivalents such as NADH and $FADH_2$, required to donate electrons to the mitochondrial electron transport chain that leads to ATP production. Glycolysis is also central to support other biosynthetic needs and branches off into the pentose phosphate pathway (PPP) to generate NADPH for redox balance and ribose-5-phosphate for nucleotide synthesis, as well as into the hexosamine biosynthesis pathway (HBP), central for N- and O-glycosylation of proteins. Besides glucose, the oxidation of fatty acids is often regarded as preferential energy source in metabolically active cells as it allows maximum ATP yield per carbon atom, in particular when oxygen is not limiting. However, the metabolic wiring of cells is not only determined by energy requirements, but also by biosynthetic needs. As the airway epithelia are heterogeneously constituted by cell subsets with different functions, it is conceivable that differentiated cells and progenitors have different metabolic requirements. Indeed, secretory cells must produce and secrete mucus components that are highly glycosylated, while multiciliated cells must sustain ciliary movement that has a high energy demand.

In this study, we show that failure to engage fatty acid metabolism during the transition from BCs to fully differentiated epithelial cells results in aberrant airway epithelium regeneration. While progenitors engage in glycolysis to sustain their proliferation, differentiation into secretory and multiciliated cells is underpinned by the engagement of fatty acid oxidation (FAO). Pharmacological or genetic impairment of FAO prevents airway epithelial differentiation and impairs the in vivo healing of airways. Moreover, our results show that differentiated cells rely on FAO to support protein glycosylation, either directly or indirectly by limiting the need for glucose to flux through the glycolytic pathway. We therefore postulate that differentiated lung epithelial cells use FAO as the most efficient form of energy generation per carbon atom in an oxygen-saturated environment while preserving glucose for other important metabolic needs, such as the fuelling of the hexosamine pathway to provide building blocks for the glycosylation needed for mucus production. These findings help to understand the physiology of lung epithelia repair and may open new therapeutic paths towards improving tissue repair after lung injury.

## Results

### Rewiring of cellular metabolism accompanies airway epithelial cell differentiation

Based on morphological, molecular and functional similarities, the mouse trachea represents an excellent model to study aspects of epithelium regeneration in human airways[4,8]. Airway progenitors isolated from adult murine tracheae can be expanded in vitro in organotypic murine tracheal epithelial cell (mTEC) cultures and induced to differentiate when subjected to an air-liquid interphase (ALI)[21]. Therefore, this culture system provides a useful model for studying epithelial regeneration in vitro, as it mimics reformation of an intact epithelium following injury.

We established surface antibody staining and flow cytometric analysis of these cultures at 10 days post ALI for identification of two cell subsets: the CD49f[+]NGFR[+] population enriched in cells expressing basal cell markers like p63 (*Trp63*) and keratin 5 (*Krt5*), and the CD49f[-]NGFR[-] subset enriched in differentiated cells, including multiciliated cells (expressing *Ccno*, *Mcidas* and *Ccdc67*) and secretory cells (positive for *Muc5ac*, *Muc5b* and secretoglobins), as assessed by quantitative PCR (Fig. 1a and Supplementary Fig. 1a). To explore the transcriptional landscape of CD49f[+]NGFR[+] cells (hereafter named basal cells) and CD49f[-]NGFR[-] cells (hereafter named differentiated cells), we performed RNA sequencing. Analysis of differentially expressed genes (DEGs) showed the regulation of pathways associated not only to the progenitor/differentiated status of the two subpopulations, but also to their metabolic status. Specifically, differentiated cells showed a strong enrichment for pathways associated with fatty acid metabolism, β-oxidation and mitochondrial activity (Fig. 1b). The gene ontology term 'fatty acid metabolism' scored as the highest among the pathways upregulated in differentiated cells, and the expression of almost all the genes in the β-oxidation pathway was elevated as compared to basal cells (Fig. 1b and Supplementary Fig. 1b). Independent screening of the 'Reactome' database by GSEA also revealed an enrichment in the 'Metabolism of lipids and lipoproteins' pathway in differentiated cells (Fig. 1c).

Functionally, differentiated cells showed higher expression of the fatty acid translocase CD36 (Fig. 1d), increased uptake of the fluorescent C16:0 fatty acid palmitate, and higher content of active mitochondria, as measured by quantification of Bodipy-palmitate, TMRM and Mitotracker Orange loading, while no significant difference was seen in the ability of cells to take up the fluorescent glucose analogue 2-NBDG (Fig. 1e). Overall, these data indicate that differentiation of airway epithelia is accompanied by adaptations in cellular metabolism towards an increased lipid uptake and usage.

To further investigate the metabolic profiles of basal and differentiated cells, we performed untargeted metabolomic and lipidomic analysis of their cellular metabolism by mass spectrometry. Differentiated cells were characterized by a substantial skewing in their lipid content as compared to basal cells (Fig. 1f), which is not driven by differences in cell number, size, and intracellular complexity, as measured by flow cytometry quantification of FSC and SSC parameters (Supplementary Fig. 1c). Upon quantile-normalization of the data to homogenize the lipid content across cell types, we assessed the differential distribution of lipid species between basal and differentiated cells (Fig. 1g, h). While phosphatidylethanolamines (PE, and related sub-species) and phosphatidylinositols (PI) showed an enrichment in differentiated cells, phosphatidylglycerols (PG) were preferentially enriched in progenitors (Fig. 1g, h). Other lipid species such as phosphatidylcholines (PC) and phosphatidylserines (PS) did not show an obviously biased distribution between progenitors and differentiated cells. These data substantiate our RNA- and flow cytometry-based

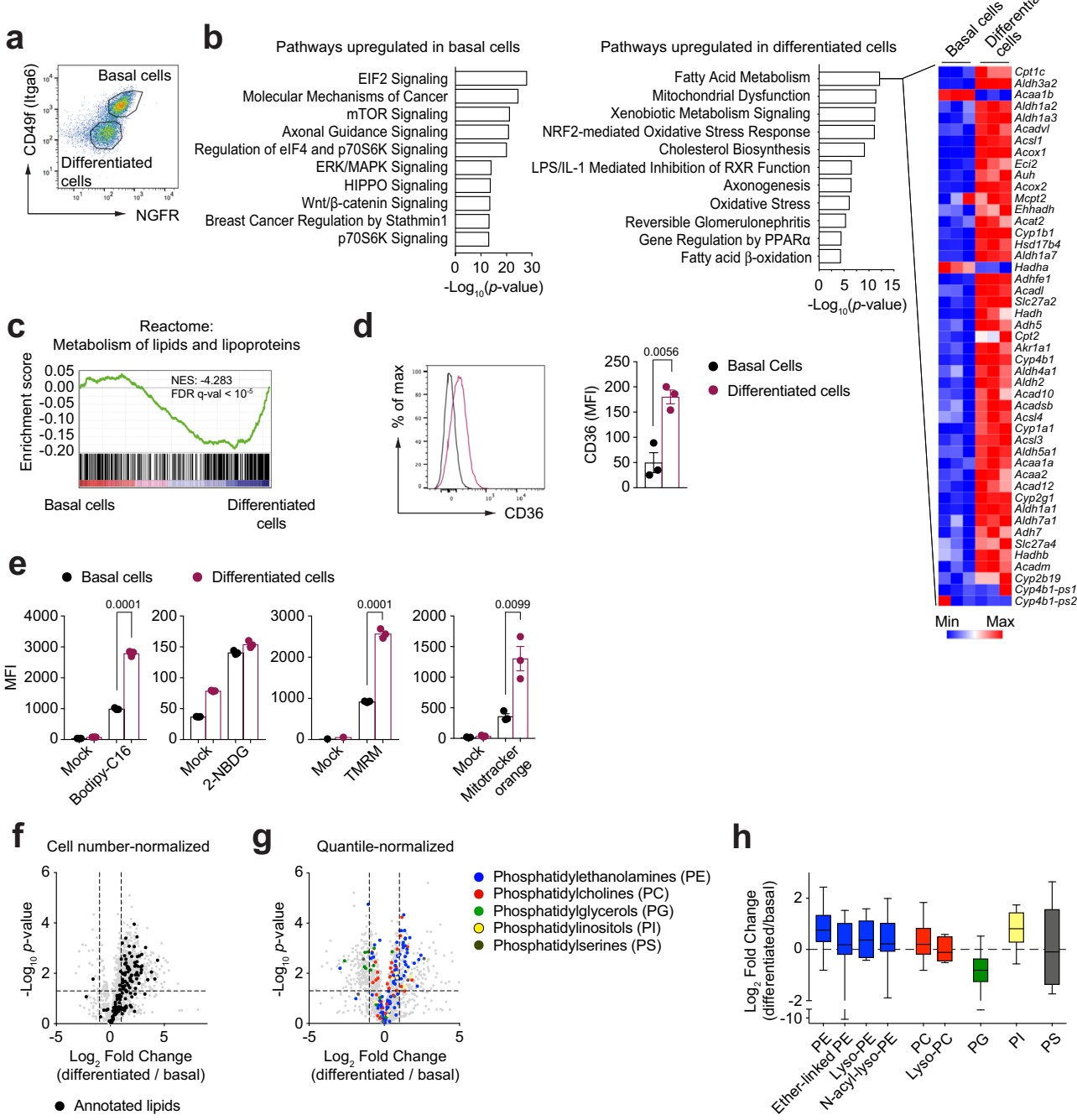

**Fig. 1 | Rewiring of cellular metabolism accompanies airway epithelial cell differentiation. a** FACS gating strategy of murine tracheal epithelial cell cultures (mTEC) collected at ALI day 10 (CD49f⁺NGFR⁺ basal cells; CD49f⁻NGFR⁻ differentiated cells). **b** Ingenuity Pathway Analysis of differentially expressed genes obtained from the RNA sequencing of basal and differentiated cells ($p$ adj <0.01), to identify differentially regulated pathways. The indicated $\text{Log}_{10}(p\text{-values})$ were calculated using the Fisher's Exact test. The heatmap showing the expression of transcripts related to the 'fatty acid metabolism' pathway was generated using the Broad Institute 'Morpheus' software. **c** Gene set enrichment analysis (GSEA) of the 'metabolism of lipids and lipoproteins' pathway. NES: normalized enrichment score; FDR: false discovery rate. **d** Flow cytometry analysis of CD36 expression on the surface of basal and differentiated cells at ALI day 10. Data show geometric mean +/− SD. Two-tailed unpaired Student's $t$ test; $n = 3$ biologically independent samples representative of 3 independent experiments. MFI: mean fluorescence intensity. **e** Flow cytometry analysis of mTEC cultures (ALI day 10) exposed to either green-fluorescent fatty acid bodipy-C16, fluorescent glucose analogue 2-NBDG, TMRM or mitotracker orange for 30 min at 37 °C. Cells were then stained to define the basal and differentiated subsets. Mean +/− S.D.; $p$ values were calculated using unpaired, two-tailed $t$ test; $n = 3$ biologically independent samples from 3 independent experiments. **f, g** Volcano plots showing the distribution of intracellular lipid species identified and quantified by mass spectrometry analysis of FACS-sorted basal and differentiated cells. Plot in **f** shows data normalized to cell number; plot in **g** shows data upon quantile normalization, to highlight distribution of lipid species between cell types. Dashed black lines indicate the fold change filter of FC > 2 and FC < −2 and the $p$-value filter of $p < 0.05$, based on two-tailed, unpaired $t$ tests, with unequal variance, FDR correction. **h** Box plots show the relative abundance of lipid species in FACS isolated basal and differentiated cells. Lines indicate median values, whiskers indicate minimum and maximum values, and boxes are limited by the 25th and 75th percentile. Plot shows data of five independent samples.

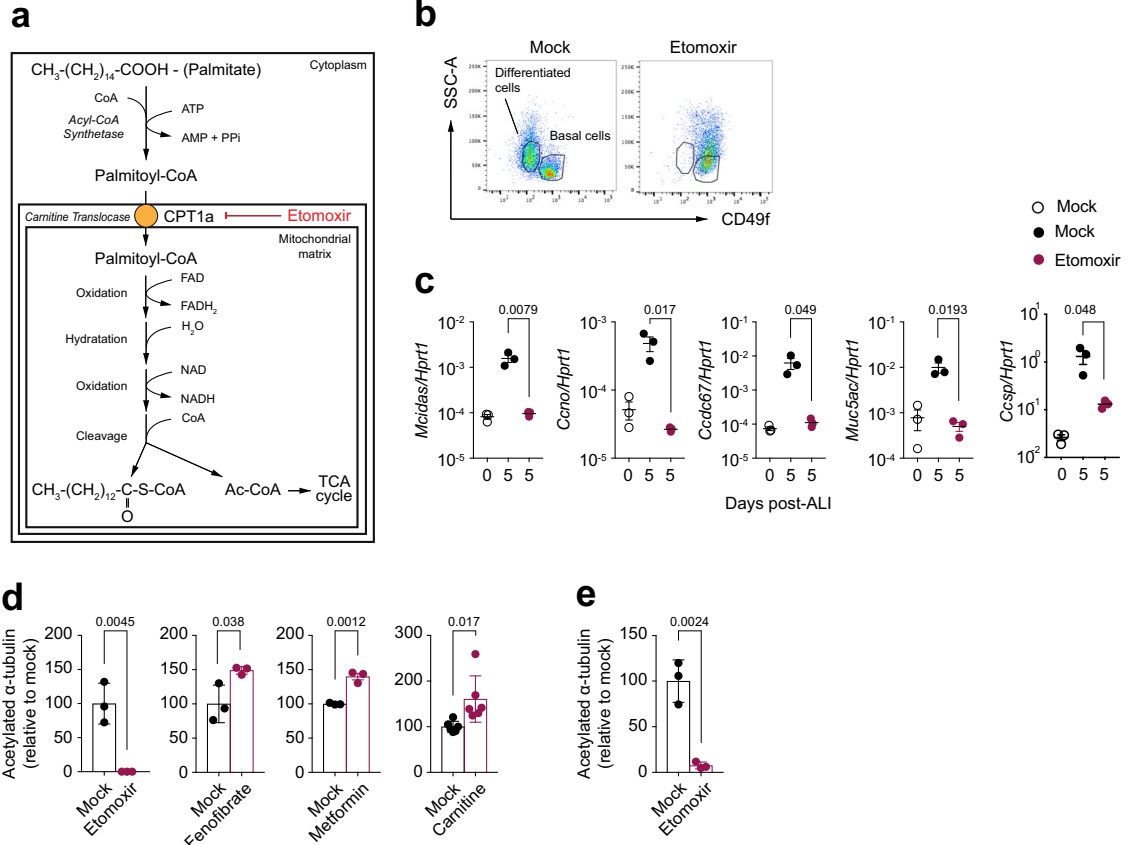

**Fig. 2 | Modulation of FAO influences epithelial cell differentiation. a** Scheme representing fatty acid β-oxidation (FAO) of palmitic acid in the mitochondria and the role of etomoxir as an irreversible inhibitor of CPT1a. **b** Flow cytometry analysis of basal and differentiated cells in mTEC cultures at ALI day 10, in presence or absence of etomoxir. Data show representative dot plots of five independent experiments. SSC: side scatter. **c** mRNA expression levels of the indicated genes in mTEC cultures at ALI day 0, or day 5 in the presence or absence of etomoxir. Values are normalized to *Hprt1*. Data show mean +/− SD and are representative of five independent experiments. Statistics were performed using two-tailed unpaired Student's *t* test; *n* = 3 biologically independent samples. **d** Fluorescence microscopy quantification of acetylated α-tubulin from entire wells of mTEC cultures, grown in

presence or absence of etomoxir, fenofibrate, metformin or carnitine, from the onset of ALI onwards. Cells were analysed at ALI day 12. Data show mean +/− SD. Statistics were performed using two-tailed unpaired Student's *t* test. *n* = 3–6 biologically independent samples of 3 independent experiments. Fluorescence intensity of mock-treated cultures was set as 100%. **e** Fluorescence microscopy quantification of acetylated α-tubulin from entire wells of human BEC cultures, grown in presence or absence of etomoxir from ALI day 9 onwards. Cells analysed at ALI day 16. Data show mean +/− SD. Statistics were performed using two-tailed unpaired Student's *t* test. *n* = 3 biologically independent samples of 3 independent experiments. Fluorescence intensity of mock-treated cultures was set as 100%.

findings and highlight the metabolic switch towards lipid metabolism that correlates with the differentiation of airway epithelial cells.

## Changes in cellular metabolism accompany differentiation of both secretory and multiciliated cells

To test whether the switch of cellular metabolism occurred in specific subsets of differentiated cells, or was a feature of the differentiation process itself, we set up a staining strategy to distinguish multiciliated, secretory and basal cells using flow cytometry. We confirmed high expression of the fatty acid translocase CD36 on both secretory and ciliated cells, following ex vivo dissociation of murine tracheae and in mTEC cultures (Supplementary Fig. 2a, b). Ciliated cells can be easily identified by intracellular markers, such as the transcriptional factor Foxj1, but this staining requires fixation and permeabilization and precludes live-cell sorting. In our staining panel, we therefore replaced intracellular markers with SiR-tubulin, a membrane-permeable live-cell dye that labels microtubules and preponderantly accumulates in microtubule-rich multiciliated cells[22]. This allowed for the identification and separation of basal, ciliated, and secretory cells from mTEC cultures. Principal component analysis (PCA) of the RNA sequencing dataset obtained from the three cell subsets confirmed the segregation of their transcriptional profiles (Supplementary Fig. 2c). Furthermore,

GSEA highlighted that genes associated to fatty acid metabolism were upregulated by both multiciliated and secretory cells, further suggesting that a switch towards fatty acid metabolism is a feature of the differentiation process rather than being characteristic of a given cell subset (Supplementary Fig. 2d).

## Pharmacological inhibition of FAO impairs airway epithelial cell differentiation

We next tested whether FAO is specifically required for differentiation. Before being oxidized in the mitochondria, fatty acids must be activated in the cytoplasm by fatty acyl-CoA synthetases. The subsequent transport of fatty-acyl-CoA molecules into the mitochondria involves an acyl-carnitine intermediate, generated by carnitine palmitoyl-transferase 1 (CPT1), an enzyme that resides in the outer mitochondrial membrane (Fig. 2a). CPT1, in concert with CPT2 and CACT (carnitine/ acyl carnitine translocase), is therefore essential to import long-chain fatty acid into the mitochondria to be funnelled into FAO[23]. We pharmacologically targeted CPT1 to assess the effect of FAO inhibition on the differentiation of the airway epithelium. In ALI-exposed mTEC cultures, the CPT1 inhibitor etomoxir prevented the differentiation of basal cell to CD49f-negative differentiated cells (Fig. 2b). Etomoxir blocked the ALI-driven upregulation of differentiated cell markers like

*Mcidas, Ccno* and *Ccdc67* (multiciliated cells), *Muc5ac* (goblet cells) and *Scgb1a1* (CCSP, secretory cells), and reduced the numbers of goblet cells developing in mTECs (Fig. 2c and Supplementary Fig. 3a). Importantly, we used etomoxir at a concentration below those shown to have off-target effects[24–26]. Indeed, we excluded that etomoxir treatment had major effects on cell viability (Supplementary Fig. 3a, b), the mitochondrial transmembrane potential (Supplementary Fig. 3c), and the generation of reactive oxygen species (ROS) in mTEC, when applied during the process of differentiation (Supplementary Fig. 3a–c), or on fully differentiated mTEC cultures (Supplementary Fig. 3d). Etomoxir treatment blocked the appearance of differentiated cells and caused a shift to SSC[high] in basal cells (Supplementary Fig. 3c), indicative of an increase in cellular granularity. Indeed, quantification of intracellular neutral lipid droplets by BODIPY 493/50 showed a significantly increased signal in CD49f[+] basal cells from etomoxir-treated cultures (Supplementary Fig. 3e), suggestive of accumulation of intracellular lipid droplets following FAO blockade. As expected, neutral lipid content was increased in differentiated cells.

Enhancing FAO by either carnitine (the major component of the carnitine shuttle), fenofibrate (a PPAR-α agonist that transcriptionally sustains FAO[27]) or metformin (that reduces acetyl-CoA carboxylase activity and increases the rate of fatty acid oxidation via AMPK activation[28]) treatments significantly increased the frequency of differentiated cells in mTEC cultures as measured by immunofluorescence staining and quantification of the ciliary component acetylated α-tubulin, while having no effects on total cell numbers (Fig. 2d and Supplementary Fig. 4a, b). A similar dependency on FAO to drive airway epithelial differentiation was observed in primary human bronchial epithelial cultures (hBEC) (Fig. 2e and Supplementary Fig. 4c), therefore expanding the implications of our findings to clinically relevant settings. Again, etomoxir treatment did not affect total hBEC cell number yet caused an accumulation of CD49f[+]NGFR[+] basal cells, in line with its inhibitory effect on cell differentiation (Supplementary Fig. 4c).

## Cpt1a deletion prevents airway epithelial cell differentiation

As previously mentioned, CPT1 constitutes a rate-limiting enzyme of FAO. To confirm genetically the role of FAO in epithelium differentiation, we set out to generate mTEC cultures from mice with the gene *Cpt1a* conditionally ablated upon treatment with tamoxifen (*R26 Cre ERT2 Cpt1a[f/f]*: *Cpt1a[−/−]*), and littermate controls (*R26 Cre ERT2 Cpt1a[+/+]*: *Cpt1a[+/−]*) (Fig. 3a). Tamoxifen treatment during the culture expansion phase resulted in excision of *Cpt1a*, with consequent reduction of its expression that was maintained for the entirety of the experiment, arguing against the emergence and outgrowth of escapees over time (Fig. 3b). No major differences were observed in mTEC cultures by light microscopy or in the total number of cells collected at different time points of ALI (Supplementary Fig. 5a). However, *Cpt1a* loss impaired the ALI-induced upregulation of differentiated cell markers like *Mcidas, Ccsp, Muc5ac* and *Muc5b* (Fig. 3b). This block in differentiation was also confirmed by the reduced number of acetylated α-tubulin-positive multiciliated cells and Muc5ac[+] goblet cells in mTEC cultures exposed to ALI for 10 days (Fig. 3c and Supplementary Fig. 5a).

Next, we assessed genome-wide transcriptional changes in *Cpt1a[+/−]* and *Cpt1a[−/−]* cultures analysed at the end of the expansion phase (ALI day 0) or at day 5 and 7 post air-exposure by RNA-seq. Cluster analysis of the transcriptional landscapes of mTEC cultures highlighted that, before the onset of ALI, undifferentiated cultures showed similar transcriptional profiles, clustering together independently of their genotypes, as indicated by the dendrogram in Fig. 3d. However, *Cpt1a*-proficient cultures exposed to ALI for 5 and 7 days diverged from both undifferentiated and *Cpt1a*-deficient cultures, with the latter maintaining a close transcriptional proximity to undifferentiated cells even in conditions that drive differentiation (Fig. 3d). Gene set enrichment analysis highlighted strong enrichment for pathways associated with both lipid metabolism and epithelial differentiation (i.e., axoneme assembly, cilium organization and movement, N-linked glycosylation) in *Cpt1a* proficient cultures (Fig. 3e). By contrast, loss of Cpt1a functions associated with the upregulation of genes involved in cell cycle and DNA replication activities (Fig. 3e). In conclusion, genetic or pharmacologic blockade of mitochondrial FAO impedes airway epithelial differentiation in vitro.

## FAO regulates distal and proximal airway epithelial cell differentiation in vivo

To assess the role of FAO in airway epithelial development in vivo, we treated *R26 CreERT2 Cpt1a[f/f]* (*Cpt1a[−/−]*) and *R26 CreERT2 Cpt1a[+/+]* (*Cpt1a[+/+]*) mice with tamoxifen to induce *Cpt1a* gene excision (Fig. 4a, b). We focussed our analysis on epithelial development and assessed the dynamics of cellular differentiation at homeostasis and in different models of airway injuries. Based on the surface expression levels of EpCam, MHCII, integrin α6 (CD49f) and the lineage marker CD24, distal lung epithelial cells can be divided into alveolar cells (EpCam[low]CD49f[low]MHCII[hi]) and small airway epithelial cells (SAEC, EpCam[hi]CD49f[hi]MHCII[low]), with the latter further subdivided into differentiated small airway epithelial cells (CD24[hi] SAEC) and CD24[low] epithelial stem/progenitor cells (CD24[low] SAEC progenitor)[29] (Gating strategy in Supplementary Fig. 6a).

At homeostasis, mouse airway epithelia are notably quiescent, and we did not observe extensive proliferation in either alveolar cells or SAEC (Supplementary Fig. 6b), or significant differences in the cellular composition of epithelia between *R26 CreERT2 Cpt1a[f/f]* and *R26 CreERT2 Cpt1a[+/+]* mice, 10 days after tamoxifen treatment (Fig. 4c, d).

To assess the dynamics of airway epithelial repair upon damage, we infected *R26 CreERT2 Cpt1a[f/f]* and *R26 CreERT2 Cpt1a[+/+]* mice with Influenza A virus (strain X31) and analysed the cellular composition of the airway epithelium 10 days post infection, at a point in time when regeneration is well underway in this infection model[30] (Fig. 4e, f). The alveolar cell and SAEC fractions were similarly represented in CPT1a-proficient and deficient animals (Fig. 4e). Intracellular Ki67 staining 10 days post-infection indicated that alveolar cells and SAEC progenitors had the highest proliferative capacity following influenza-induced damage, indicating active epithelial tissue regeneration (Supplementary Fig. 6b). Importantly, epithelial cell proliferation was unaffected following *Cpt1a* deletion (Supplementary Fig. 6b). However, consistent with our in vitro findings, a greater accumulation of progenitors at the expense of differentiated SAEC was evident in CPT1a-deficient mice (Fig. 4f), suggesting an impaired transition from progenitors to differentiated cell.

Due to the nature of this transgenic model, tamoxifen-induced deletion of *Cpt1a* could occur in any cell type and at any stage of development, potentially impacting different biological aspects of the airway epithelium and other cells, from their developmental potential to the functionality of mature cells. To verify that the observed phenotype associated to the non-hematopoietic compartment independently of CPT1a-driven immune functions, we generated bone marrow chimeric mice by adoptive transfer of bone marrow cells from *R26 CreERT2 Cpt1a[f/f]* or *R26 CreERT2 Cpt1a[+/+]* mice into lethally irradiated *R26 CreERT2 Cpt1a[f/f]* or *R26 CreERT2 Cpt1a[+/+]* recipients. Chimeric mice were then treated with tamoxifen to induce *Cpt1a* deletion and infected with Influenza A virus (Fig. 4g). Flow cytometry analysis of lungs collected 10 days post-infection confirmed accumulation of progenitors at the expense of differentiated SAEC in animals carrying the deletion of *Cpt1a* in the non-hematopoietic compartment, independently of the genetic status of the donor-derived immune cells (Fig. 4h).

Unlike the intestines and the haematopoietic system, the lung epithelium is not maintained solely by professional stem cells

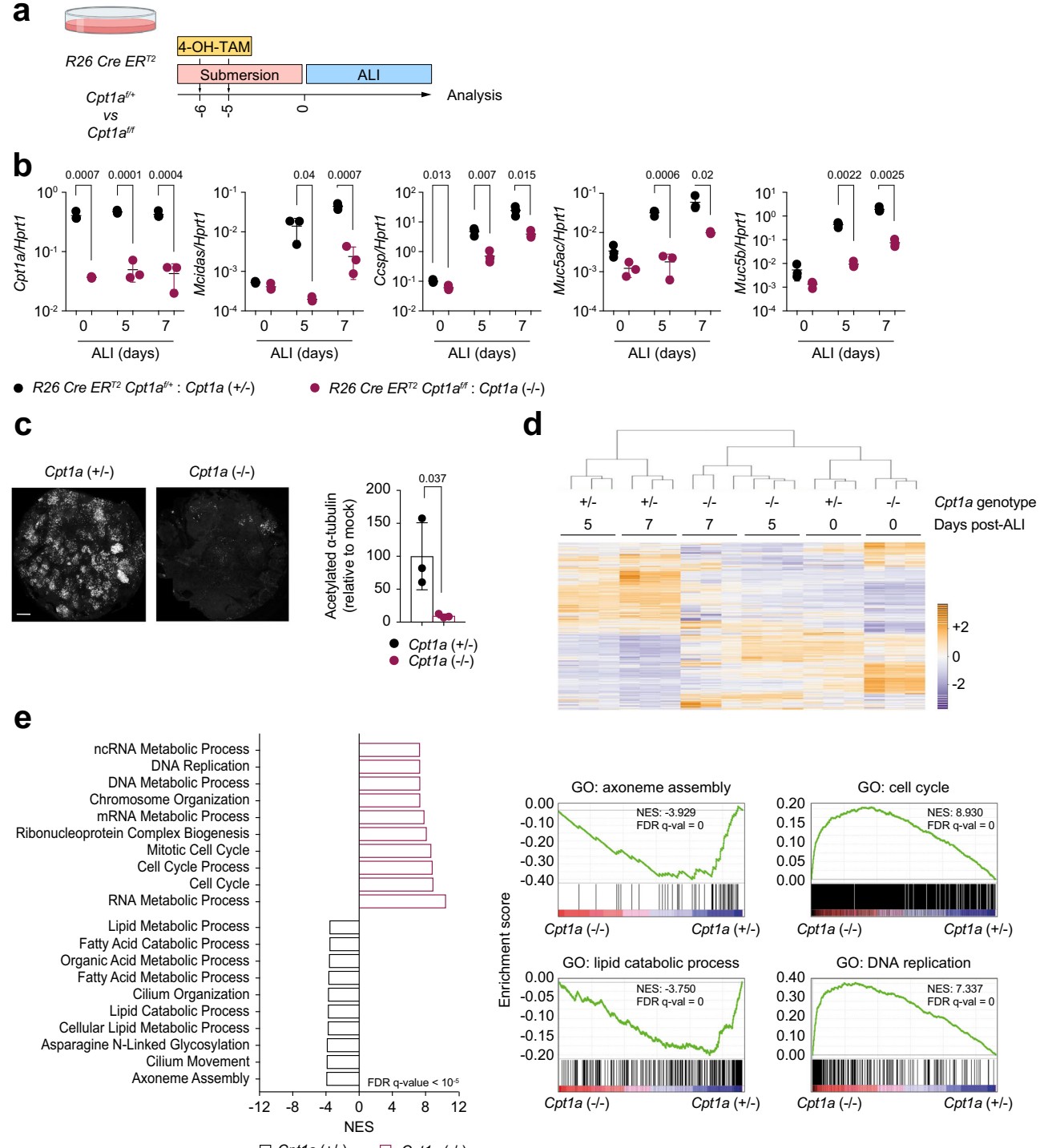

**Fig. 3 | Cpt1a deletion prevents airway epithelial cell differentiation. a** Scheme representing the experimental strategy used throughout this figure. *R26 CreERT2 Cpt1a^f/f^* and *R26 CreERT2 Cpt1a^f/+^* mTEC cultures were treated with *4-hydroxytamoxifen* (4-OH-TAM) for two days during the proliferation phase to induce Cre recombinase activity. Cultures were analysed at ALI day 0, 5 and 7. **b** qPCR analysis of mRNA expression levels of the indicated genes in *Cpt1a^+/-^* and *Cpt1a^-/-^* mTEC cultures at ALI day 0, 5 and 7. Values are normalized to *Hprt1* expression. Data show mean +/− SD. Statistics were performed using multiple unpaired *t* tests and adjusted using the Holm−Sidak correction. *n* = 3 biologically independent samples. **c** Fluorescence microscopy analysis and quantification of acetylated α-tubulin from *Cpt1a^+/-^* and *Cpt1a^-/-^* mTEC cultures. Cells were analysed at ALI day 10. Data show mean +/− SD. Statistics were performed using two-tailed unpaired Student's *t* test. *n* = 3 biologically independent samples. Fluorescence intensity of mock-treated cultures was set as 100%. Scale bar: 1 mm. **d** Unsupervised hierarchical clustering and heatmaps of DEGs from RNA sequencing of *Cpt1a^+/-^* and *Cpt1a^-/-^* mTEC cultures analysed at ALI day 0, 5 and 7. Data are cumulative of three biological replicates. Colour coding is as per legend and indicates the row Z-score. **e** GSEA of pre-ranked genes in *Cpt1a^+/-^* and *Cpt1a^-/-^* mTEC cultures analysed at ALI day 5. All pathways shown have FDR < 10^−5^. NES normalized enrichment score, FDR false discovery rate, GO gene ontology.

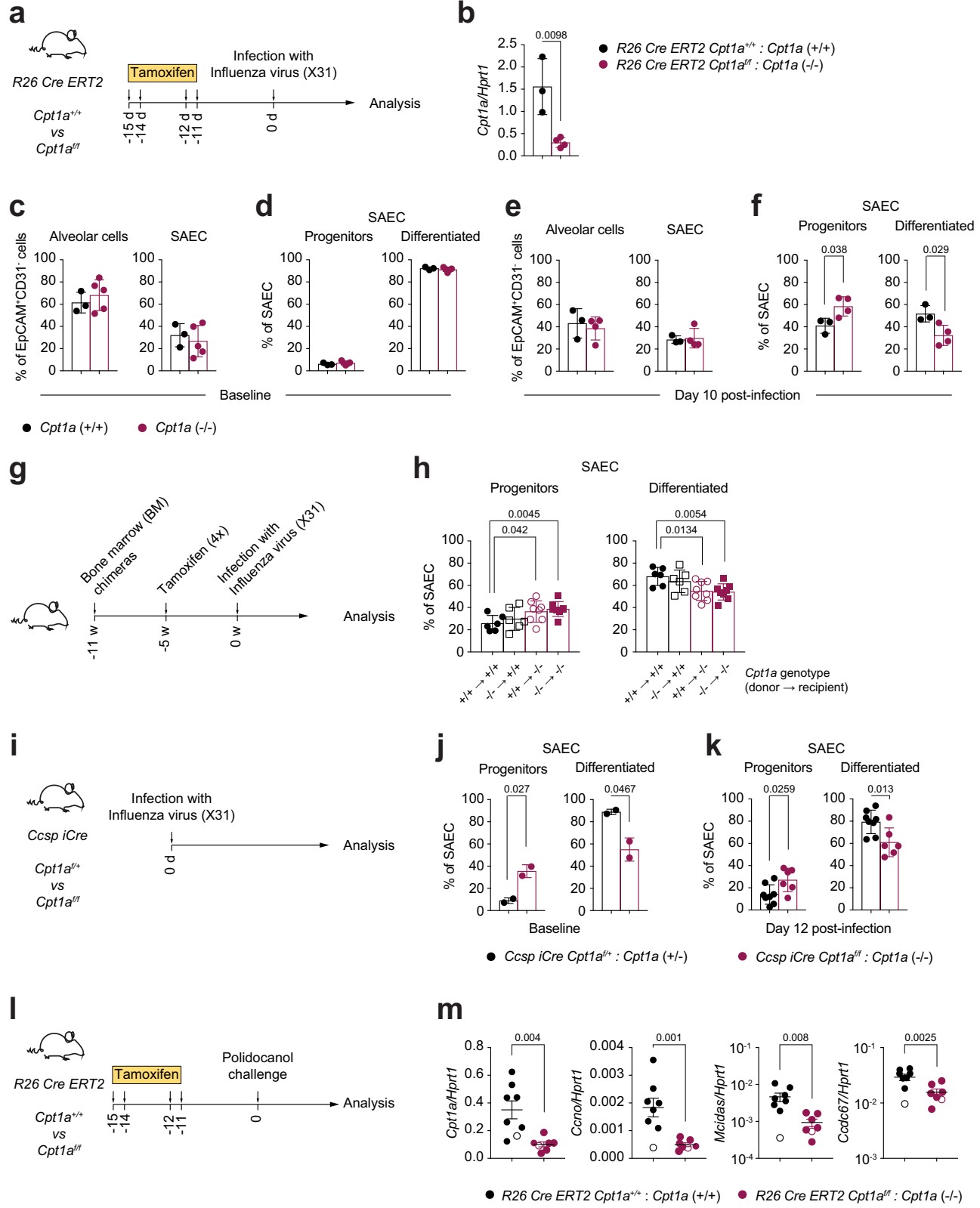

differentiating through a well-delineated stem/progenitor/differentiated cell hierarchy. Rather, depending on the extent of tissue damage, various cell types are recruited to participate in the repair process by re-entering the cell cycle and differentiating into multiple lineages[5,7]. Recently, a distinct subpopulation of club-like lineage-negative epithelial cells (expressing the gene *Scgb1a1*, also known as *Ccsp*) has been identified as stem/progenitor cells in the airway epithelium[31,32]. Based on this observation, we crossed *Cpt1a*[fl/fl] mice to mice carrying the codon improved Cre recombinase gene under the control of the *Ccsp* locus to generate *Ccsp iCre Cpt1a*[f/f] and control *Ccsp iCre Cpt1a*[f/+] littermates (Fig. 4i).

We first tested this model in vitro by generating mTEC cultures and found that *Ccsp*-driven *Cpt1a* deletion significantly impaired the development of differentiated cell (Supplementary Fig. 6c). Therefore,

**Fig. 4 | FAO regulates distal and proximal airway epithelial cell differentiation in vivo. a** Experimental strategy used in (**a**–**f**). *R26 CreERT2 Cpt1a*$^{+/+}$ (*Cpt1a*$^{+/+}$) and *R26 CreERT2 Cpt1a*$^{f/f}$ (*Cpt1a*$^{-/-}$) mice were treated with tamoxifen and infected with Influenza A virus (strain X31). Each dot represents an individual mouse, $n = 3$–5 (**c**, **d**) or $n = 3$–4 (**e**, **f**). **b** mRNA expression levels of *Cpt1a* in lungs from *R26 CreERT2 Cpt1a*$^{+/+}$ and *R26 CreERT2 Cpt1a*$^{f/f}$ mice, 15 days after tamoxifen treatment. Values are normalized to *Hprt1* expression. Data show mean +/− SD. **c**–**f** Flow cytometry analysis of lungs from *Cpt1a*$^{+/+}$ and *Cpt1a*$^{-/-}$ mice on day 0 (**c**, **d**), or day 10 post-infection (**e**, **f**), to assess the fraction of alveolar type II cells (CD45$^-$ CD31$^-$ EpCam$^{low}$ CD49f$^{low}$ MHCII$^{hi}$), small airway epithelial cells (SAEC, CD45$^-$ CD31$^-$ EpCam$^{hi}$ CD49f$^{hi}$ MHCII$^{low}$) (**c**, **e**), progenitor (CD24$^{low}$) and differentiated (CD24$^{hi}$) SAEC (**d**, **f**), as defined in Supplementary Fig. 6a. Data show mean +/− SD. **g** Experimental strategy used in (**h**). Bone marrow chimeric mice were generated as indicated, using *R26 CreERT2 Cpt1a*$^{+/+}$ and *R26 CreERT2 Cpt1a*$^{f/f}$ mice as donors or recipients. Chimeric mice were then treated with tamoxifen and infected with X31. $n = 6$–8. **h** Flow cytometry analysis of lungs of chimeric mice on day 10 post-infection. Data show mean +/− SD. **i** Scheme of the experiment shown in (**j**) and (**k**). *Ccsp iCre Cpt1a*$^{f/+}$ (*Cpt1a*$^{+/-}$) and *Ccsp iCre Cpt1a*$^{f/f}$ (*Cpt1a*$^{-/-}$) mice were infected with X31. **j**, **k** Flow cytometry analysis of lungs isolated from the indicated mice on day 0 (**j**) $n = 2$, or day 12 post-infection (**k**) $n = 6$–8. Data show mean +/− SD. **l** Experimental strategy. *R26 CreERT2 Cpt1a*$^{+/+}$ and *R26 CreERT2 Cpt1a*$^{f/f}$ mice were treated with tamoxifen to induce *Cpt1a* gene excision and challenged with polidocanol. **m** mRNA expression levels of the indicated genes in tracheae isolated from *Cpt1a*$^{+/+}$ and *Cpt1a*$^{-/-}$ mice, 10 days after polidocanol treatment. Values are normalized to *Hprt1* expression. Data show mean +/− SD. Empty symbols represent baseline gene expression. Statistics in **b**, **f**, **h**, **j**, **k**, and **m** were performed using two-tailed unpaired Student's *t* tests.

---

we used these mice to assess the dynamics of airway epithelial cell differentiation in vivo, both at homeostasis and following influenza-induced airway injury. Interestingly, the epithelia of *Ccsp iCre Cpt1a*$^{f/f}$ mice showed an imbalance in the distribution between progenitors and differentiated SAEC as compared to *Ccsp iCre Cpt1a*$^{f/+}$ littermates, not only after infection but already at steady state, with an accumulation of progenitors at the expense of differentiated SAEC (Fig. 4j, k). These data indicate that even when restricted to mature *Ccsp*$^+$ secretory cells and cells derived from the aforementioned *Ccsp*$^+$ progenitors, *Cpt1a* deletion was responsible for aberrations in the cellular composition of airway epithelia, and that during cell differentiation a metabolic switch to FAO is required for homeostasis of the normal epithelium as well as its physiological regeneration after injury.

To further assess the requirement of FAO during epithelial repair, we employed an alternative model of airway damage using the detergent polidocanol which, when given in a small volume, affects only the proximal airways, including trachea. Polidocanol has been shown to temporarily remove airway epithelial cells in mice and to induce the proliferation of normally quiescent basal cells in the proximal airways[33]. As shown in Supplementary Fig. 7, intranasal instillation of the detergent caused widespread removal of the epithelium, with loss of the superficial columnar layer and exposure of the denuded basal lamina. At 6 days post injury, repair was well underway, with reappearance of an undifferentiated columnar layer and, by day 14, re-establishment of a normal-appearing epithelium. To test the role of FAO in proximal airway epithelial differentiation, we administered polidocanol intranasally to both *R26 CreERT2 Cpt1a*$^{f/f}$ and *R26 CreERT2 Cpt1a*$^{+/+}$ mice (Fig. 4l). RNA analysis of the upper airways 10 days after polidocanol treatment showed that *Cpt1a* deficiency in *R26 CreERT2 Cpt1a*$^{f/f}$ mice impeded the upregulation of genes required for the de novo generation of multiciliated cells (Fig. 4m). Immunostainings on the whole tracheae confirmed that at 14 days post treatment, the trachea basal laminae of both genotypes were no longer exposed and had been repopulated by a layer of cells (Supplementary Fig. 8, DAPI staining), suggesting that the differences in epithelial repair do not depend on differences in the proliferative potential of the two genotypes. However, *Cpt1a*-deficient mice showed a reduction in acetylated α-tubulin and CCSP staining at the epithelial surface, suggesting a defect in the differentiation of both multiciliated and secretory cells. Of note, tamoxifen treatment did not induce uniform deletion of *Cpt1a* across the whole epithelial layer. Indeed, the tracheae of tamoxifen-treated *R26 CreERT2 Cpt1a*$^{f/f}$ mice were characterized by areas where CPT1a protein was still detectable (Region B, Non-deleter) adjacent to areas devoid of CPT1a expression (Region A, Deleter). Importantly, this 'internal control' further highlighted the correlation between the lack of CPT1a and the absence of multiciliated cells labelled by acetylated α-tubulin staining, as well as secretory cells labelled by CCSP (Supplementary Fig. 8).

In conclusion, our data show that genetic impairment of FAO profoundly affects the physiological regeneration of proximal and distal airway epithelia upon injury, by preventing the transition from basal to differentiated cells.

## FAO feeds the hexosamine biosynthesis pathway to sustain protein glycosylation in differentiated airway epithelial cells

FAO is intertwined with several metabolic pathways. The main product of the oxidation of fatty acids is acetyl-CoA, which, among other fates, is fed into the TCA cycle to sustain oxidative phosphorylation and ATP production[34]. Nucleo-cytosolic levels of acetyl-CoA have been linked to the epigenetic control of gene expression in multiple cell types, and fatty acid-derived acetyl-CoA primarily drives the lymphoangiogenic commitment of endothelial cells through histone acetylation of a specific set of genes[15]. To investigate which cellular process in developing epithelia is supported by FAO, we first tested whether the cytoplasmic acetyl-CoA pool was maintained by FAO[35], and whether it played a role in the fate commitment of progenitors to differentiated cells. We generated mTEC cultures in the presence or absence of etomoxir and tested whether the acetyl-CoA precursors acetate and citrate were able to bypass the FAO inhibition to support airway epithelial differentiation. The pH of cell cultures was adjusted to avoid excessive acidity of the cultures. Neither acetate or citrate could rescue the inhibition of epithelial differentiation by etomoxir, as measured by the expression of *Mcidas* and *Muc5ac* (Fig. 5a). Of note, acetate and citrate alone prevented epithelial differentiation (Fig. 5a), possibly by increasing acetyl-CoA levels that provided negative feedback to FAO[36], or through increasing levels of malonyl-CoA that negatively feeds back to CPT1 activity[37]. In addition, inhibiting two enzymes required for the generation of the cytosolic acetyl-CoA pool, namely the mitochondrial citrate transporter Slc25a1 by BTC, or the ATP citrate lyase (Acly) by SB204990, did not affect the generation of multiciliated cells (Fig. 5b).

As ATP production through FAO and oxidative phosphorylation is highly efficient when oxygen is not limiting, we also explored the alternative hypothesis that FAO may be primarily required for ATP production. However, cellular ATP levels were unaffected by FAO blockade both in developing mTEC cultures or in fully differentiated airway epithelia (Fig. 5c).

Our RNA sequencing analysis of mTEC cultures showed that the pathway of *N*-glycosylation is negatively affected by *Cpt1a* deletion (Fig. 3e). *N*-glycosylation and mucin-type *O*-glycosylation both require UDP-N-acetyl glucosamine (UDP-GlcNAc), whose synthesis is mediated by the hexosamine biosynthesis pathway, fed by glucose, glutamine, acetyl-CoA and uridine as substrates (Fig. 5d). RNA sequencing analysis showed that almost all the enzymes involved in the HBP and many mannosyltransferases are upregulated upon airway epithelial differentiation (Fig. 5d, e). Moreover, metabolomic analysis of basal and differentiated cells showed that uracil and uridine levels were strongly increased in differentiated cells relative to basal cells, while the levels of other nucleobases, nucleosides, or nucleotides were not significantly different (Fig. 5f).

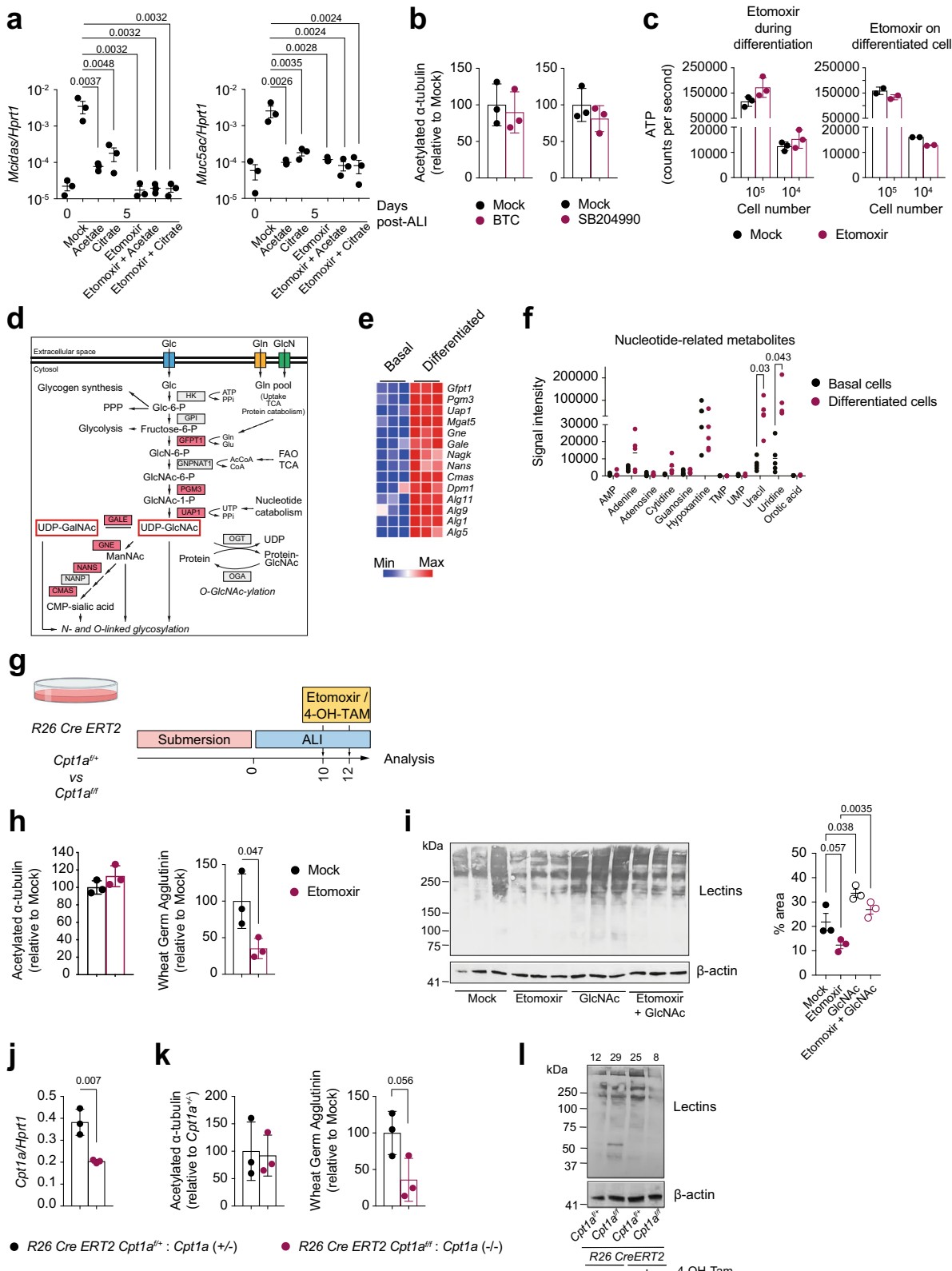

To test the impact of FAO inhibition on *N*-glycosylation, we treated *fully differentiated* mTEC cultures with etomoxir and assessed global protein glycosylation by lectin staining (Fig. 5g). While etomoxir treatment of fully differentiated cultures did not affect the number of acetylated α-tubulin-positive multiciliated cells, it did substantially reduce global protein glycosylation levels, as measured by staining with wheat germ agglutinin (Fig. 5h).

Furthermore, western blot analysis of cellular lysates showed reduced protein glycosylation in etomoxir-treated cultures that could be rescued by N-acetyl glucosamine (GlcNAc) supplementation (Fig. 5i). In addition, genetic deletion of *Cpt1a* in fully differentiated mTEC cultures of *R26 CreERT2 Cpt1a^{f/f}* mice (Fig. 5g), showed that late *Cpt1a* deletion did not affect the number of differentiated cells, but impacted on the levels of glycosylated

**Fig. 5 | FAO is not required to maintain cytosolic acetyl-CoA and ATP levels but to sustain protein glycosylation in differentiated airway epithelial cells.**
**a** mRNA levels of the indicated genes in mTEC cultures at ALI day 0, or day 5 in presence or absence of the indicated compounds. Data show mean +/− SD. One-way ANOVA with Tukey's multiple comparison tests; $n = 3$. **b** Quantification of acetylated α-tubulin staining of mTEC cultures, grown in presence or absence of BTC or SB204990, at ALI day 10. Mean +/− SD, two-tailed unpaired Student's $t$ test, $n = 3$. Fluorescence intensity of mock-treated cultures set as 100%. **c** Intracellular ATP levels in mTEC cultures either mock or etomoxir-treated during the differentiation phase (ALI day 0–6) or once differentiated (ALI day 10–12). Data show mean +/− SD, $n = 3$. **d** Scheme of the hexosamine biosynthetic pathway. Red highlights genes upregulated in differentiated cells. **e** Heatmap of genes linked to HBP differentially expressed in differentiated and basal cells. **f** Relative abundance of the indicated metabolites in basal and differentiated cells, measured by mass spectrometry. Dot plots show means of $n = 5$ independent samples; statistics performed

using multiple $t$ tests. **g** Experimental strategy used throughout (**h**) and (**l**). Wild-type mTEC cultures (**h**, **i**), or mTEC from *R26 CreERT2 Cpt1a^f/f^* and *R26 CreERT2 Cpt1a^f/+^* mice (**j–l**) were treated with etomoxir (**h**, **i**), or 4-OH-TAM (**j–l**) for 48 h once differentiated (ALI day 10–12) and analysed at ALI day 12. **h** Quantification of acetylated α-tubulin and wheat germ agglutinin staining. Mean +/− SD, two-tailed unpaired Student's $t$ test, $n = 3$. Fluorescence intensity of mock-treated cultures set as 100%. **i** Protein glycosylation levels and quantification in lysates of differentiated mTEC cultures, after 48 h treatment with the indicated compounds. Mean +/− SD, two-tailed unpaired Student's $t$ test, $n = 3$. **j–l** *R26 CreERT2 Cpt1a^f/f^* and *R26 CreERT2 Cpt1a^f/+^* cultures were differentiated for 10 days in ALI and treated with 4-OH-TAM. Cells were analysed for *Cpt1a* expression (**j**), acetylated α-tubulin and lectin staining (**k**) and glycosylation levels of whole lysates (**l**). **j**, **k** Mean +/− SD, two-tailed unpaired Student's $t$ test, $n = 3$. Numbers in (**l**) represent densitometric quantification of glycosylation levels.

proteins (Fig. 5j–l), even though in this condition the deletion of the gene was only about 50% efficient (Fig. 5j).

Taken together, these data suggest that during differentiation of airway epithelia, FAO is dispensable as a source of acetyl-CoA for histone acetylation or to fuel ATP production, but required to sustain the hexosamine biosynthesis pathway, thus promoting *N*-glycosylation and mucin-type *O*-glycosylation of proteins in differentiated epithelial cells. In this scenario, limiting the metabolite supply for *N*-glycosylation by blocking FAO and forcing glycolysis[38] would interfere with the differentiation of basal cells and/or alter the cellular functions of mature secretory and multiciliated cells.

## Discussion

Here, we show that the differentiation of airway epithelial cell progenitors towards both multiciliated and secretory cells during lung regeneration requires a metabolic shift towards fatty acid oxidation (FAO). Using pharmacological and genetic tools in vitro and in vivo, we show that a block of FAO impairs the regeneration of a fully functional airway epithelium, and that enhancing FAO promotes epithelial repair.

Cellular metabolism is emerging as a key regulator of stem cell quiescence and proliferation[39,40], and switching between different metabolic pathways is crucial for the activation of quiescent stem cell populations and the onset of differentiation[41–43]. In tissues, stem cells are activated and mount acute, regenerative responses that occur upon sensing of activating signals, including growth factors and inflammatory cytokines[14,44]. This proliferative phase needs to be transient to prevent aberrant repair and organ dysfunction, and it is key to understand how changes in the epigenetic landscape and gene expression required for cell-fate specification are regulated in time[45–47]. Environmental factors such as nutrient availability may also play an important role in this process. For instance, the availability of oxygen may determine the balance between aerobic glycolysis and oxidative phosphorylation (OXPHOS), as described in hematopoietic stem cells[48]. It is also becoming clear that metabolic remodelling is not only a feature of early development but also a major hallmark in post-injury tissue regeneration; studying this phenomenon may therefore help our understanding of how metabolic remodelling takes place in pathological situations, how it impacts disease progression and resolution, and how it could be modulated to restore tissue homeostasis.

In this study, we utilised the well-established organotypic model of ALI cultures of mouse tracheal epithelial cells to investigate the mechanisms of differentiation following airway epithelium damage[21]. This model consists of two distinct stages. Initially, progenitors proliferate to form a continuous, impermeable, monolayer of cells linked by tight junctions. Next, the apical side of the monolayer is exposed to air, while the basal side remains in contact with the culture medium, triggering the differentiation of progenitor cells into multiciliated, secretory and goblet cells. This progression is accompanied by dramatic changes in the transcriptional profile of epithelial cells, and the

pathways induced indicate strong metabolic adaptation of cells to the luminal environment of the airways aimed at sustaining the newly acquired cellular functions of differentiated cells.

The progression from basal to differentiated epithelial cells is underpinned by a switch from glycolysis, that supports the need for biomass of the highly proliferative progenitors, towards FAO. While differentiated cells are equipped to take up fatty acids from the environment, it is conceivable that substrates for FAO may as well derive from fatty acid synthesis, in a 'futile cycle' previously shown to occur in memory T cells[49]. Pharmacological inhibition of FAO with etomoxir, and genetic deletion of the FAO rate-limiting enzyme *Cpt1a*, confirmed the central role of FAO in controlling the differentiation of airway epithelial cells. These observations were reproduced in human bronchial epithelial cultures, expanding the implications of our findings to clinically relevant target cells. In clinical settings, several lung conditions including COPD and ARDS have been successfully treated with L-carnitine and other agents promoting FAO, but the underlying mechanism is controversial (reviewed in ref. [50]). In addition, animal models of emphysema and acute lung injury showed protective effects of enhancing FAO by carnitine or fenofibrate treatment[51,52]. Our data help explain the mechanistic underpinnings of these treatments as they indicate that pharmacological promotion of FAO improved or accelerated re-differentiation of lung epithelia upon injury.

The engagement of FAO by differentiated cells lining the airway lumen is likely sustained by the ready availability of oxygen. While exposure to air (ALI) may be the key step, it is interesting to speculate that oxygen sensing via HIF prolyl hydroxylases may control FAO engagement. It has indeed been previously shown that HIF1α actively represses FAO[53], thus oxygen sensing may trigger HIF1α degradation and release FAO inhibition. It would therefore be interesting to explore how hypoxia regulates the metabolic choices of airway epithelial cells and their fate commitment. Local lung hypoxia, as caused by influenza virus, has been shown to determine the fate decision of cells during alveolar regeneration[54], and this concept could be well extended to the airway compartment. It is also possible that transcription factors involved in the differentiation of airway epithelial cells, such as the aryl hydrocarbon receptor[55], may have a role in coordinating the metabolic commitment of epithelial cells[56].

In contrast to airway epithelial cells, there is a substantial body of work regarding alveolar epithelia metabolism. At steady state, primary mouse alveolar epithelial cells utilize both glycolysis and FAO for their metabolic needs[52]. However, in a mouse model of bleomycin-induced alveolar injury, autophagy rewires cellular metabolism by supporting glycolysis and the pentose phosphate pathway (PPP), both essential for AT2 cell proliferation[18].

In addition, glutamate metabolism is also important for AT2 proliferation, and reductions in the levels of the enzymes driving glutamine catabolism were observed in AT2 cells in both IPF patients and bleomycin-treated mice, perhaps explaining the aberrant profibrotic

responses in these conditions[19]. Data from metabolite analysis in patients with severe IPF also suggest that mitochondrial transport is impaired and fatty acids accumulate in IPF lungs[57]. Since mitochondrial fatty acid import is a vital prerequisite for β-oxidation, the authors concluded that FAO is downregulated in the IPF lung tissues.

In airway epithelia, while the main cell type driving recovery post injury are basal cells[5], other cell types can also contribute to repair. In murine distal airway epithelium, which (in contrast to its human counterpart) lacks basal cells[5], other precursors have been described, including Club cells, a subset of which selectively survives naphthalene treatment and serves as progenitor for regeneration in this injury model. Similar to the above findings for alveolar epithelia, both autophagy and glycolysis were shown to be important for proliferation, but not differentiation of these precursors, in a murine model of asthma[20].

Our data show that FAO blockade leads to impaired development of both secretory and multiciliated cells, two cell types that may have overlapping but also distinct metabolic requirements for functioning correctly. Considering the high energetic demands of multiciliated cells to sustain cilia beating, and of secretory cells to support protein synthesis, we were surprised that FAO inhibition did not affect the ATP pool in differentiated cells. However, FAO is intimately linked to several metabolic pathways, and it is conceivable that glycolysis could compensate the inhibition of FAO to maintain viable ATP levels in differentiated cells. We thus hypothesized that the employment of FAO in differentiated cells could spare carbon equivalents flowing from glycolysis into the TCA cycle, to eventually feed into alternative metabolic pathways. Indeed, our RNA sequencing data highlighted that differentiated cells upregulated several genes involved in the control of the hexosamine biosynthesis pathway, an offshoot of glycolysis. The HBP generates substrates for N-glycosylation and mucin-like O-glycosylation of proteins, well-described post-translational modifications of proteins produced by secretory and goblet cells[58]. In line with the transcriptional data, we found that FAO inhibition in differentiated cultures impaired glycosylation of proteins, while leaving the differentiation status of the cultures unaffected. We hypothesize that FAO sustains glycosylation by sparing glycolysis from feeding into the TCA cycle for energy production, and thereby redirecting carbons into the HBP[38]. Alternatively, FAO may serve as a primary source to provide acetyl-CoA for the HBP. This metabolic rewiring serves airway epithelial cells to maintain elevated levels of protein glycosylation, required to generate the structure of the mucus layer that protects the host from infectious and chemical insults[58]. Moreover, previous evidence highlighted a role for glycosylation in mediating cell migration and coordinating wound healing during repair of airway epithelia[59,60]. It is not uncommon that carbons entering glycolysis are repurposed for biosynthetic needs. Early studies in rat lungs suggested glucose sparing from use in glycolysis in favour of the PPP, most likely to generate NADPH to protect from oxidative stress[61]. While in these historic studies cell-type specific resolution is not achieved, it is an interesting speculation that the most air-exposed cells, including differentiated airway epithelial cells, might switch to FAO to reserve glucose to provide antioxidant protection via NADPH production. Similarly, during T cell proliferation, glucose-derived carbons are funnelled into the pentose phosphate shunt to sustain the de novo generation of nucleotides for DNA synthesis[62].

Our work highlights the metabolic rewiring underlying the differentiation of airway epithelial cells and opens several questions that remain to be investigated. ALI-exposed mTEC cultures are heterogeneous in terms of differentiated cell types. While inhibition of FAO uniformly prevented the development of all subsets of differentiated cells in airway epithelia, the precise chart of subset-specific metabolism is yet unclear. The broad role for FAO in controlling the differentiation process is also still unclear. Our data argue against a requirement for epigenetic regulation of the differentiation process itself, but the issue needs further investigation.

The Notch signalling pathway has emerged as a critical regulator for the development, homeostasis, and regeneration of the respiratory system[63]. Although the precise control of Notch signalling in regeneration remains elusive, Notch activation was proven crucial for the differentiation into secretory cells and dispensable for the self-renewal of basal cells at homeostasis and during repair[64]. Both Notch receptors and their ligands are glycoproteins and their sugar modifications play important regulatory roles at various steps of the pathway activation, including receptor folding, trafficking and ligand interactions (reviewed in ref. [65]). It is therefore conceivable that the reduction in protein glycosylation we described under conditions of FAO blockade could influence cellular differentiation through modification of Notch signalling, although this question will require a detailed study.

To conclude, we show here that metabolic rewiring is an essential component of airway epithelial differentiation and suggest that FAO serves to free glucose from feeding energy production and to redirect carbon from the glycolytic pathway into the HBP, essential for the high glycosylation demands in airway epithelia. Our results improve the understanding of epithelial biology and provide new targets for metabolic intervention to promote repair and re-differentiation post lung insult.

## Methods

### Ethics statement
Our research complies with all relevant ethical regulations as specified below.

### Mice
All animal experiments were approved by the institute ethics committee and the Home Office, UK, under project license P9C468066, and carried out in accordance with the Animals (Scientific Procedures) Act 1986. All experiments used male and female mice at 8–12 weeks of age bred at the Francis Crick Institute under specific pathogen-free conditions. All genotypes were bred on a C57BL/6J background. $Cpt1a^{fl/fl}$ ($Cpt1a^{tm1.1Pec}$)[16] were crossed to mice carrying a tamoxifen-inducible CreERT2 in the ROSA26 locus ($Gt(ROSA)26Sor^{tm1(cre/ERT2)Thl}$)[66]; or mice carrying iCre recombinase in the Scgb1a1 (Ccsp) locus ($Scgb1a1^{tm1(icre)Fjd}$)[67]. To induce Cre recombinase activity, 2 mg of tamoxifen (20 mg/ml in corn oil; Sigma-Aldrich) was administered i.p. daily for 4 days. To generate bone marrow chimeras, single-cell suspensions from femurs of donor mice were depleted of erythrocytes using ACK lysis buffer (155 mM $NH_4Cl$, 10 mM $KHCO_3$, 100 μM EDTA). $5 \times 10^6$ cells were injected i.v. into recipient mice (as indicated in figures), which had been irradiated with $2 \times 6.5$ Gy, 15-h interval using a $^{137}Cs$ source. Mice were given water supplemented with 0.02% enrofloxacin (Baytril; Bayer Healthcare) for 4 weeks after transplantation and used for experiments 8 weeks after reconstitution.

### Influenza viruses
X31 stocks were grown in the allantoic cavity of 10 day-embryonated hen's eggs and were free of bacterial, mycoplasma and endotoxin contamination. All viruses were stored at −80 °C and titrated on Madin−Darby canine kidney (MDCK) cells to determine tissue culture infective dose ($TCID_{50}$), according to the Spearman-Karber method.

### Infections
Mice were infected with X31 (10,000 $TCID_{50}$ in 30 μl PBS), under light anaesthesia (3% isoflurane) intranasally. Mice were infected at 8 to 12 weeks of age, except for chimeric mice that were infected at 18 to 20 weeks of age.

## Induction of epithelial damage by polidocanol

20 µl of 2% polidocanol in phosphate-buffered saline (PBS) was applied intranasally to anesthetized mice. To assess the damage qualitatively, animals were killed by barbiturate overdose at the indicated time point after instillation and their tracheae were removed for conventional histology and RNA extraction.

## Primary airway epithelial cultures and in vitro treatments

Isolation and culture of primary murine mTECs were performed as previously described[21]. In brief, tracheal cells isolated by enzymatic treatment were expanded in a T-75 flask to 100% confluence with the Rho kinase inhibitor Y27632 (antibodies-online.com) at 10 µM. Cells were then trypsinised and seeded ($3 \times 10^4$ cells/transwell) onto 0.4 µm pore size clear PET membranes (Greiner) coated with a collagen solution. Cells were grown in submersion until confluent, and then exposed to air to establish ALI. CPT1a deletion was induced by treatment with 400 nM 4-OH-tamoxifen for 2 days during the expansion phase (in submersion) or on fully differentiated cells (10 days after air exposure).

Human biological samples were sourced ethically, and their research use was in accord with the terms of the informed consents. Primary human bronchial epithelial cells were purchased from Lonza and cultured as per manufacturer's instructions. In brief, cells were expanded in a T-75 flask and then harvested for seeding onto transwells (Greiner) at $3 \times 10^4$ cells per insert. At confluence, liquid was removed from the upper chamber to establish ALI.

Drug treatments of both murine and human TECs were started at the onset of ALI and maintained for the duration of the experiment, unless otherwise stated. Concentrations of drugs (all from Sigma-Aldrich) were as follows: etomoxir 50 µM, fenofibrate 1 µM, carnitine 250 µM, metformin 200 µM, BTC 2 mM, SB204990 60 µM, acetate 5 mM, citrate 5 mM.

Uptake of fatty acid and glucose was measured by exposure of cells to 0.5 µM fluorescent Bodipy-C16 (4,4-Difluoro-5,7-Dimethyl-4-Bora-3a,4a-Diaza-s-Indacene-3-Hexadecanoic Acid) or to 20 µM 2-NBDG (Invitrogen) for 30 min at 37 °C, 5% $CO_2$. Labelling of mitochondria was done by exposure of cells to the cell permeant Mito-tracker Orange (Molecular Probes) (100 nM) or to TMRM (50 nM) for 30 min at 37 °C, 5% $CO_2$. Cells were then washed 3x in PBS/2% sera, trypsinised and analysed by flow cytometry.

BODIPY 493/503 staining of neutral lipids for flow cytometry was performed by incubating cells in a 2 µM BODIPY 493/503 solution in PBS, for 30 min at 37 °C, 5% $CO_2$. Cells were then stained as desired to define the different subpopulations.

In all in vitro experiments with mTEC and hBEC cultures, each dot represents an independent culture, with $n \geq 3$.

## RNA isolation

mTEC cultures were lysed in RLT + β-mercaptoethanol and RNA isolated using the Qiagen RNeasy mini kit, according to the manufacturer's instructions. 300 ng total RNA was reverse-transcribed using the qPCRBIO cDNA synthesis kit as per manufacturer's instructions. RT-qPCR was performed on an Applied Biosystems Quantstudio 3 RT-qPCR machine with 1x qPCRBIO Probe Mix Lo-ROX, and 1x Taqman primers. Results were normalized to the housekeeping gene *Hprt1*, and shown as $2^{-\Delta Ct}$. The following probes (Applied Biosystems) have been used: *Hprt1* (Mm00446968_m1), *Ccno* (Mm01297259_m1), *Mcidas* (Mm01308202_m1), *Ccdc67* (Mm00725262_m1), *Muc5ac* (Mm01276718_m1), *Muc5b* (Mm00466391_m1), *Scb1a1* (Mm00442046_m1), *p63* (Mm00495793_m1), *Krt5* (Mm01305291_g1), *Cpt1a* (Mm01231183_m1).

## RNA sequencing

RNA-sequencing was performed on the HiSeq 4000 system (Illumina) with Single End 75 bp reads. Read quality trimming and adaptor removal was carried out using Trimmomatic (version 0.36). Reads were aligned to the mouse genome (Ensembl GRCm38 release 89) using STAR (version 2.5.2a) and gene level counts were obtained using the RSEM package (version 1.2.31). Differential expression analysis was carried out with DESeq2 package (version 1.20.0) within R version 3.5.1. Genes were considered to be differential expressed with $p$ adj $\leq 0.05$. Gene Set Enrichment analysis (GSEA, version 2.2.3) was performed using gene lists ranked using the Wald statistic. Gene set pre-ranked analysis was carried out using C2 canonical pathways v5.2 and C5 biological processes v5.2. All parameters were kept as default except for enrichment statistic that was changed to classic and the max size which was changed to 500,000. Gene signatures were considered significant if FDR-value $\leq 0.05$. Ingenuity Pathway Analysis was performed using differentially expressed genes ($p$ adj < 0.01). The indicated log($p$-values) were calculated using the Fisher's Exact test.

## Lung and trachea dissociation and flow cytometry

For cell isolation from lung tissues, mice were euthanised (600 mg/kg pentobarbital/17 mg/kg mepivacaine) and then perfused with 10 ml of ice-cold PBS through the right ventricle of the heart. 1.5 ml Dispase II (Roche) (5 mg/ml in IMDM) was then injected intratracheally into the lungs, followed by 0.4 ml 1% low-gelling agarose solution (in PBS) (Sigma). Mice were then placed on ice allowing the agarose/dispase-filled lungs to set. Lungs were then dissected and placed in 2 ml Dispase II solution for 30 min to dissociate epithelial cells. Lungs were passed through a 100 µm strainer, before a 10-min DNase I digestion (50 µg/ml) (Sigma). Following digestion, lung homogenates were passed through a 70 µm strainer and centrifuged at 1400 r.p.m. for 5 min at 4 °C, before red blood cell lysis. Single-cell suspensions were preincubated with anti-FcgRIII/II (Fc block), before a 30-min incubation with the indicated fluorochrome-labelled antibodies. For Ki67 staining, cells were fixed and permeabilised before staining. Cells were analysed using a Fortessa X20 (Becton Dickinson).

Airway epithelial cells from ex vivo tracheae were dissociated using Dispase II (Roche) (5 mg/ml in IMDM), for 2 h at 37 °C. After incubation, dissociated tissues were washed in media containing 10% FCS, passed through a 70 µm strainer, pelleted and stained with the indicated antibody panel for flow cytometry.

## ROS staining

Reactive oxygen intermediates were detected by flow cytometry, after exposure of the cells to $H_2DCFDA$ (2′,7′-dichlorodihydrofluorescein diacetate, Biotium) 10 µM, for 30 min at 37 °C, 5% $CO_2$.

## SiR-tubulin staining

SiR-Tubulin (Spirochrome/Cytoskeleton Inc.) is a live cell dye which stains microtubules, a major structural component of cilia. mTEC cultures were incubated in ALI media supplemented with 1 µM SiR-tubulin and 10 µM verapamil, a broad-spectrum efflux pump inhibitor, for 1 h at 37 °C [68]. Cultures were then washed in PBS, trypsinised to generate single-cell suspension and stained with the indicated antibodies for flow cytometric analysis and sorting.

## Histology and immunofluorescence

Dissected tracheae were fixed overnight in 4% paraformaldehyde (PFA), then embedded in paraffin and sectioned. For immunofluorescent staining of both paraformaldehyde-fixed mTEC and dewaxed trachea specimens, samples were permeabilized with 0.1% Triton X-100 for 15 min at room temperature and then blocked in 0.5% bovine serum albumin for 1 h. Primary antibodies specific for acetylated α-tubulin (T7451, Sigma), CPT1a (Proteintech), CCSP and Muc5ac (Abcam) and Foxj1 (eBioscience) or wheat germ agglutinin (WGA-AF 555, 3 µg/ml) were added in 0.5% bovine serum albumin and incubated for 1 h at room temperature. After washing with PBS, fluorochrome-conjugated secondary antibodies (Alexa Fluor, Life Technologies) were added for 1 h at room temperature. Finally, slides were washed in

PBS and mounted using Vectashield mounting medium with DAPI (Vector labs). Images were acquired on an Olympus VS120 slide scanner. Measurements of acetylated α-tubulin and lectin staining were made using regions of interest that cover the whole transwell. Binary images were generated by global thresholding and measured using the ImageJ 1.4j application.

### Metabolomic and lipidomic analysis

Untargeted metabolomic and lipidomic analysis of progenitor and differentiated cells was performed as follows. After sorting of CD49f⁺NGFR⁺ and CD49f⁻NGFR⁻ cells, $3.5 \times 10^5$ cells were washed in ice-cold 3% glycerol and the pellet resuspended in 1 volume ice-cold 100% methanol. Samples ($n = 5$ independent cultures) were immediately added with 1 volume of ice-cold ultrapure water and thoroughly vortexed. Samples were further added with 1 volume ice-cold chloroform and vortexed for 5 min on ice. Samples were centrifuged for 3 min, $20,000 \times g$, at 4 °C. The top aqueous phase was collected in a new tube, whereas the bottom organic phase was collected in glass vial. Samples were finally dried using a Genevac EZ2 speed vac and stored at −80 °C until further processing following protocols previously described[69].

LC-QTOF-MS analysis of metabolites and lipids was performed as previously described[69]. For both measurements an Agilent 1290 Infinity II UHPLC in line with a Bruker Impact II QTOF-MS operating in negative ion mode has been used. Mass calibration was performed at the beginning of each run.

For polar metabolites, the scan range was from 20 to 1050 Da. LC separation was on a Phenomenex Luna propylamine column (50 × 2 mm, 3 μm particles) using a solvent gradient of 100% buffer B (5 mM ammonium carbonate in 90% acetonitrile) to 90% buffer A (10 mM NH4 in water). Flow rate was from 1000 to 750 μL/min. Autosampler temperature was 5 degrees and injection volume was 2 μl.

For lipids, the scan range was from 50 to 1600 Da. LC separation was on a Zorbax Eclipse plus C18 column (100 × 2 mm, 1.8 μm particles) using a solvent gradient of 70% buffer A (10 mM ammonium formiate in 60:40 acetonitrile:water) to 97% buffer B (10 mM ammonium formiate in 90:10 2-propanol:acetonitrile). Flow rate was 400 μl/min, autosampler temperature was 5 °C and injection volume was 2 μl.

Metabolites and lipids were analysed and identified by fragmentation and retention time using Metaboscape software (version 2021b, Bruker). Statistical analysis of untargeted metabolomic data was performed applying a Student's $t$ test between metabolites identified in basal cells and metabolites identified in differentiated cells. Data pre-processing including feature extraction, feature deconvolution, and metabolite/lipid identification was performed using Metaboscape software (version 2021b, Bruker). During this step, known background ions were removed and only those features that were on average at least 2-fold above process blank were retained. For annotation metabolite/lipids must match 2 or more of the following criteria: (1) less than 3 mDa mass deviation, (2) MS/MS Spectrum match score of more than 600, (3) less than 24 s retention time deviation relative to a reference list of retention times of standard compounds. In addition, all annotated metabolites/lipids were manually checked for plausible retention times. To investigate differences in lipid composition, we quantile normalized the complete lipidomics dataset using R function "normalize.quantiles()" from package preprocessCore.

### Western blot

Tissues and cells were lysed in RIPA buffer containing proteinase and phosphatase inhibitors. Equal amounts of whole lysates were electrophoresed on TGX precast gels (Bio-Rad), transferred to nitrocellulose membrane (Whatman) and incubated with primary antibodies at 4 °C overnight. The following primary antibody was used: anti-β-actin (4970 s, Cell Signaling Technology, 13E5, 1:5000). The immunoblot was developed using horseradish peroxidase-conjugate secondary antibodies (Bio-Rad) and the ECL Western Blotting Detection kit, as instructed by the manufacturer (GE Healthcare).

For detection of glycoproteins, membranes were first block by incubation in Glyco-Free Blocking Solution (Bioworld) for 30 min at room temperature and then incubated in PBS containing 2 μg/ml biotinylated lectin (MAA/MAL I + II, GlycoMatrix), for 30 min at room temperature. After washing with PBS/0.05% Tween-20, membranes were stained with the Vectastain ABC-AP kit (Vector Laboratories), as per manufacturer's instructions, using BCIP/NBT as alkaline phosphatase substrate.

### Intracellular ATP measurements

Total ATP levels were measured using the ATPlite kit (Perkin Elmer).

### Statistics and reproducibility

Unless otherwise stated, the statistical significance of differences among groups was evaluated with a two-tailed, unpaired Student's $t$-test using GraphPad Prism. Differences were considered statistically significant at $p < 0.05$. In some experiments, statistics were performed using multiple $t$ tests adjusted using the Holm–Sidak correction. The statistical test used, and the significance are always reported in the figure legends. All the presented data have been independently repeated at least three times.

### Reporting summary

Further information on research design is available in the Nature Portfolio Reporting Summary linked to this article.

## Data availability

Sequencing data are available in GEO under accession codes GSE209686 and GSE218663. Source data are provided with this paper.

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

## Acknowledgements

We thank the Crick Flow cytometry, Advanced sequencing, Bioinformatics, Histopathology and Animal facilities for excellent support. This work was funded by the Francis Crick Institute, which receives its core funding from Cancer Research UK (FC002085), the UK Medical Research Council (FC002085), and the Wellcome Trust (FC002085). P.C. is supported by Grants from Methusalem funding (Flemish government), the Fund for Scientific Research-Flanders (FWO-Vlaanderen) and the European Research Council (ERC) Advanced Research Grant (EU- ERC74307).

## Author contributions

S.C., M.V. and A.W. conceived the idea and designed the experimental strategy. S.C., M.V., J.M., K.F. and J.B. designed and performed experiments and analysed and interpreted data. M.L. performed bioinformatics analysis. P.C. made key experimental tools available. S.C., M.V. and A.W. wrote the manuscript. All authors edited the manuscript. All authors read and approved the final manuscript.

## Funding

## Competing interests

The authors declare no competing interests.
