## [Peer Review File · Nature Communications]

Repair of airway epithelia requires metabolic rewiring towards fatty acid oxidationReviewer #1 (Remarks to the Author):

Authors adopted in vitro ALI cultures and in vivo infection mouse model to study effects of fatty acid metabolism in airway epithelial differentiation. By using metabolic inhibitors and Cpt1a KO mice, they proposed that FAO is important in basal cell differentiation. However, it is not clear which lineage differentiation (ciliated cells or goblet cells) requires for FAO. Authors overlooked several important studies in the field. Findings in this study are interesting but this study is preliminary because they do not provide further mechanistic information underlying this finding.

Authors are confused in using some terms. Respiratory epithelia is usually referred to alveolar gas-exchanging epithelia and a short part of distal conducting airway epithelia (only in humans, not in mice). Focus of this study seems to be tracheal basal cells. In addition, basal cells are normally considered as stem cells in trachea and intra-lobar conducting airways that give rise to club cells. Club cells are progenitor cells and are able to differentiate into ciliated cells and goblet cells. Authors call basal cells as progenitor cells, which is not clear and may lead to confusion in the field. It is better to call basal cells.

Authors described current understanding of changes metabolism of immune cells and endothelial cells during lung injury. They stated that cellular metabolism was not studied in lung epithelia after injury, but this is absolutely wrong. Glucose metabolism and fatty acid metabolism have been studied in club and AT2 cells after lung injury (PMID:31748541; PMID:32059792; PMID:35625656). Actually, studies on lung metabolism and lung disease have been recently reviewed (PMID:30485759; PMID:33864479). However, all of these important literature that are more relevant to current manuscript were overlooked in the introduction or discussion. This leads to a disconnection of this work to the existing understanding of metabolic regulations of lung epithelial stem progenitor cells.

Line 59: It is not fair to state in that way. Lung epithelial stem/progenitor cell hierarchy has actually been well defined now, please refer to a review of PMID:28780147.

Line 117: Why does the mouse trachea represent an excellent model to study aspects of epithelium regeneration in human small airways? It is a counterpart of human large airways.

Figure 1. Differentiated cells in CD49f-NGFR- population include club cells (progenitor cells), ciliated cells and probably goblet cells (terminally differentiated cells). Pathways upregulated in differentiated cells (right panel in 1b) do not distinguish these three cell types. In vivo study in Figure 4 does not provide this information either. Basic question would be that ciliated differentiation, goblet cell differentiation or both require FAO. In addition, what the expression level of CD36 in club cells, ciliated cells or goblet cells as CD36 was demonstrated expressed in AT2 cells (PMID:29029397). CD36 expression in pooled differentiated cells does not provide sufficient information on the role of FAO in basal cell function. This information can be collected by IF staining of CD36 and CCSP, ACT, clca3 in mouse airways respectively.

Figure 2. Differentiated cells disappear in presence of Etomoxir (2b), but a large portion of basal cells shifted to SSC-hi, suggesting that this drug promotes intracellular vesicles. What happens to basal cells in presence of Etomoxir? In addition to q-PCR data, IF staining was also required to confirm for each lineage differentiation, including Muc5Ac and CCSP (2c). IF images must be shown in 2d. In 2e, human BEC cells are bronchial epithelial cells, not relevant to tracheal basal cells which is not helpful.

Figure 3, IF staining of CCSP and mucins are also required (3c). And nuclei staining must be also provided in ACT staining. Naphthalene is known to kill club cells, and ozone to kill ciliated cells. Naphthalene-induced or ozone-induced lung injury models should be used for Cpt1a KO mice to confirm in vitro findings on the regulatory role of Cpt1a on

proximal airways.

Figure 4. influenza infection has been demonstrated to induce dedifferentiation of club cells into basal cells (PMID:24196716). Did authors have a similar finding? And influenza infection-induced pathologies restrict to distal airways and alveoli. If authors are interested in regulation of alveolar AT2 cell differentiation by FAO, organoid cultures with FAO inhibitors must be conducted.

There are several epithelial progenitor cells in small airways including variant club cells, bronchioalveolar stem cells, etc. what do SAECs stand for in this aspect? What are differentiated cells then do authors refer to here? IF staining of in vitro cultures of in vivo experiments must be conducted to verify.

Extended figure 4, please also show tracheal and alveolar epithelia after polidocanol. If this injury model is not specific to certain epithelial cells in the lung, it will be difficult to evaluate functions of their mother cells?

If alveolar part was included in the study, "airway" in the title of this manuscript seems to be inappropriate.

Reviewer #2 (Remarks to the Author):

In this manuscript, Dr. Crotta and team report their findings on the importance of a metabolic shift from glycolysis to fatty acid oxidation (FAO) during repair from airway epithelial cell injury. Specifically, they find that FAO is necessary for maintaining mitochondrial respiration while allowing glucose to be preferentially used in the hexosamine biosynthesis pathway, supporting protein glycosylation and epithelial cell differentiation during repair.

Overall, I found this to be a very well written manuscript reporting a detailed series of experiments that complement each other to form a comprehensive picture of the role of FAO in this pathology. The results appear to be consistent across several different models (robust to both the injury mechanism and the method of FAO modulation). The conclusions are highly consistent with the data presented. In addition, the area of research is important, as injury prevention is often not possible by the time of clinical presentation for many acute lung pathologies, so mechanisms to improve healing/repair are an important contribution.

Major Comments:

My only major comment concerns the discussion. This section reads more like a summary of the results rather than a broader discussion of study relevance, importance, and comparison to existing literature on airway repair and metabolic response to lung injury. This strategy feels redundant, especially given the excellent summary statements presented in the results section, which keep the reader well caught-up with the pertinent findings even before reaching the discussion section. It seems that at most 1-2 paragraphs of the discussion should be needed to summarize the key findings, allowing the rest of the discussion to be converted to a broader conversation about how these findings fit with what is known and the potential novel implications for advancing the field of airway repair after acute lung injury.

Minor Comments:

- 1) Can the authors provide additional information on the mass spectrometry portion of the metabolic assay? Sample prep and data processing are well covered but I could not find information on the assay itself.**
- 2) Can the authors discuss their choice to perform untargeted metabolomics rather than targeted (or even better fully quantitative) metabolite measurements? Particularly if the focus was on FAO there should be a limited number of metabolites that could be measured with high validity.**
- 3) I could not find a statistical analysis section that covered the metabolomic analyses (or really that covered any of the data presented other than the RNA sequencing).**

Further details would be helpful.

Reviewer #3 (Remarks to the Author):

This is a comprehensive, rigorous manuscript describing a heretofore not well-defined impact of fatty acid oxidation (FAO) on airway epithelial cell fate in mice. The approach of the authors was to compare the metabolic profiles of airway stem cells (mostly basal cells) with their differentiated progeny. They discovered clear differences in the lipid profiles and then undertook numerous mechanistic studies both in vitro and in vivo that came to the conclusion that FAO is required for differentiation of progenitors to differentiated airway cells, especially ciliated cells. I find the conclusion and supporting data convincing and my comments mostly minor. The limitation of this study is mainly that the authors had to go to extremes to uncover a phenotype and a convincing roll for FAO in airway cell differentiation. The roll of FAO emerged only after they completely blocked import of palmitoyl-CoA into mitochondria by deleting or pharmacologically blocking the rate limiting mitochondrial transporter of palmitoyl-CoA (cpt1a). One wonders whether this scenario would ever occur in humans without congenital enzyme deficiency. Nonetheless the data set are high quality, new to the field, and provide a clear take home message.

Specific criticisms:

1. The argument that the lung is low in glucose is mis-stated in the introduction. The provided reference refers only to the thin airway surface liquid layer (ASL). This should be stated specifically because otherwise there is no evidence or reason to believe glucose levels are low in the lung. It is true that most of the cells in the lung are mainly using respiration and not glycolysis for energy but this appears to be due mostly to the relatively high oxygen levels and not low glucose.

2. Fig 2e. The data show dramatic decrease in alpha-tubulin/ciliated cells but there are not controls to assure little toxicity with these treated human BEC cultures. The number of basal cells in mock and treated conditions should also be shown. In line with other cells studied here, one would expect more progenitors (mainly basal cells) in the treated cultures. This is a relevant point because in other settings blockade of FAO can lead to apoptosis in the lung (Yao et al, Am J Respir Cell Mol Biol 60:667, 2019.)

3. Extended Fig 4D. The cpt1a immunofluorescence is barely visible in the cpt1a -/- images of Region B even though there is acetylated tubulin in some cells. One cannot tell whether the cells expressing acetylated tubulin are the cells without detectable cpt1a (-/-), as in other data, or if there is haploinsufficiency in some cells from partial recombination and that is sufficient to promote differentiation ?

The uncertainty raised by this point gets to the relevance of all the data as it translates to normal or diseased lung function in humans. It seems unlikely that complete loss of cpt1a function ever occurs in humans without congenital deficiency. However, hypoxia does frequently occur and would attenuate FAO. Is this degree of FAO suppression sufficient to affect airway basal cell to ciliated cell differentiation, perhaps partially ?

Point-by-point rebuttal

We thank all the Reviewers for their thoughtful comments that have helped improve this study greatly. Concerns have been addressed as outlined in the point-by-point response below. New data can be either found in the point-by-point reply, or they have been integrated in the new version of our manuscript as indicated. Our responses to the reviewers' comments are in blue.

REVIEWER COMMENTS

Reviewer #1 (Remarks to the Author):

Authors adopted in vitro ALI cultures and in vivo infection mouse model to study effects of fatty acid metabolism in airway epithelial differentiation. By using metabolic inhibitors and Cpt1a KO mice, they proposed that FAO is important in basal cell differentiation. However, it is not clear which lineage differentiation (ciliated cells or goblet cells) requires for FAO. Authors overlooked several important studies in the field. Findings in this study are interesting but this study is preliminary because they do not provide further mechanistic information underlying this finding.

Thanks to Reviewer 1's comments, we have improved our flow cytometric separation of differentiated cells and now provide information regarding the use of FAO in specific subsets of differentiated cells (outlined in Methods line 650-656). Briefly, we show in the novel Supplementary Figure 2 that the differentiation process itself required engagement of FAO, and that both multiciliated cells and secretory cells show upregulated FAO, as compared to basal cells. This is true in vitro and in vivo and extends from CD36 expression to pathway analysis of RNAseq performed on sorted secretory and multiciliated cells in comparison to basal cells.

Considering the additional data now included in this point-by-point reply and throughout the revised version of manuscript, we believe our manuscript now provides higher depth to the mechanistic insights already mentioned in the original version of the manuscript.

Authors are confused in using some terms. Respiratory epithelia is usually referred to alveolar gas-exchanging epithelia and a short part of distal conducting airway epithelia (only in humans, not in mice). Focus of this study seems to be tracheal basal cells. In addition, basal cells are normally considered as stem cells in trachea and intra-lobar conducting airways that give rise to club cells. Club cells are progenitor cells and are able to differentiate into ciliated cells and goblet cells. Authors call basal cells as progenitor cells, which is not clear and may lead to confusion in the field. It is better to call basal cells.

These are very good points raised by Reviewer 1. We now amended the manuscript to reflect the suggestion of the Reviewer. We avoid the term respiratory epithelium and use the term airway epithelium throughout. Where appropriate (i.e. in mTEC cultures and ex vivo trachea analysis) we use the term basal cells, instead of progenitors. We do use the term progenitors where we do whole lung analysis.

Authors described current understanding of changes metabolism of immune cells and endothelial cells during lung injury. They stated that cellular metabolism was not studied in lung epithelia after injury, but this is absolutely wrong. Glucose metabolism and fatty acid

metabolism have been studied in club and AT2 cells after lung injures (PMID:31748541; PMID:32059792; PMID:35625656).

We thank the referee for pointing out this omission and agree that metabolism has been studied in AT2 and Club cells. We have now added a section referring to this and have quoted the relevant literature (line 82-87, 464-483). In reference to PMID 35625656 quoted above, our studies on mTECs, the pharmacological blockade of glutaminolysis did not interfere with differentiation and did not alter total cell numbers, arguing against a major role of glutamine metabolism in airway epithelial regeneration (see graph below). We would like to point out here already that the subject of this study is airway epithelial repair, not alveolar epithelial repair. Therefore, the metabolic and repair potential of AT2 cells is not subject of this study.

Pharmacological block of glutaminolysis does not have any influence on either proliferation or differentiation of mTEC cultures
(a) Fluorescence microscopy quantification of acetylated α -tubulin in mTEC cultures that were either mock treated or treated with BPTES 5 μ M from the onset of ALI onwards. Cells were analyzed at ALI day 10. Data show mean \pm SD and. Statistics were performed using two-tailed unpaired Student's *t* test. Fluorescence intensity of mock-treated cultures was set as 100%. Cell count from parallel cultures is also shown.

Actually, studies on lung metabolism and lung diseased have been recently reviewed (PMID:30485759; PMID:33864479). However, all of these important literature that are more relevant to current manuscript were overlooked in the introduction or discussion. This leads to a disconnection of this work to the existing understanding of metabolic regulations of lung epithelial stem progenitor cells.

We have added one of these important reviews to the introduction of the paper (line 62), while the other one is specific to autophagy which is not our subject here. We have also attempted to be more precise in the introduction and discussion (464-483), and introduced new relevant references.

Line 59: It is not fair to state in that way. Lung epithelial stem/progenitor cell hierarchy has actually been well defined now, please refer to a review of PMID:28780147.

We agree with the referee that our statement is unclear. We have now modified the text in line 59-64 to clarify what we mean, i.e. that precursor – progenitor relationships in lung epithelia are less linear and more varied than in e.g. hemopoiesis or gut epithelia, and that the ability of cells to undergo transdifferentiation and dedifferentiation adds to the complexity of these relationships. We have added both the above reference and the more recent review by Basil et al., Cell Stem Cell 2020, which clearly details the wide variety of lung cells that can contribute to epithelial repair, depending on the type, location and severity of the damage incurred.

Line 117: Why does the mouse trachea represent an excellent model to study aspects of epithelium regeneration in human small airways? It is a counterpart of human large airways.

We base this statement on the classic review articles by Hogan and colleagues (e.g. 2010) and by more recent literature, such as Zepp and Morrissey (2019), which point out that mouse tracheae have a cellular composition that is similar to human tracheae, large and small airways.

This contrasts with small airways in mice which are largely missing basal cells (Zepp and Morrisey 2019).

The counterparts of mouse trachea are therefore both large and small human airways. Specifically, in mice, the pseudostratified epithelium is largely restricted to the trachea, and it transitions to a simple columnar epithelium devoid of basal cells already in the bronchi. In humans, pseudostratified epithelium containing basal cells extends to terminal bronchioles (0.5 mm diameter) (Hogan Review 2010). It is therefore correct and largely backed up by the current literature as reviewed in the above-cited publications that mouse tracheae are representative of an extended part of human airways.

Figure 1. Differentiated cells in CD49f-NGFR- population include club cells (progenitor cells), ciliated cells and probably goblet cells (terminally differentiated cells). Pathways upregulated in differentiated cells (right panel in 1b) do not distinguish these three cell types. In vivo study in Figure 4 does not provide this information either. Basic question would be that ciliated differentiation, goblet cell differentiation or both require FAO.

In addition, what the expression level of CD36 in club cells, ciliated cells or goblet cells as CD36 was demonstrated expressed in AT2 cells (PMID:29029397). CD36 expression in pooled differentiated cells does not provide sufficient information on the role of FAO in basal cell function. This information can be collected by IF staining of CD36 and CCSP, ACT, clca3 in mouse airways respectively.

Reviewer 1 raises a valid point. As the Reviewer points out, we can identify basal cells as CD49f⁺NGFR⁺ and can therefore make definitive statements on CD36 expression in these cells. As the distribution of the surface expression of CD36 is not binary but gradual, differences are relative and not “black-and-white”. We therefore chose flow cytometry rather than immunofluorescence staining on lung sections to obtain quantitative data on relative differences as required by Reviewer 1. To refine the analysis in differentiated cells, we have now established a flow cytometry staining protocol that allowed us to distinguish basal cells, multiciliated cells (MCCs) and secretory cells. The results in the new Supplementary Figure 2 show increased CD36 expression in both MCCs and secretory cells as compared to basal cells. Along the same line, RNA sequencing data of basal cells, MCCs and secretory cells, sorted based on the new staining protocol, validated our original hypothesis that FAO is required during the differentiation process itself. Basal cells, MCCs and secretory cells segregated in discrete areas of the PCA space (Supplementary Figure 2c), highlighting their different transcriptional profiles. However, MCCs and secretory cells did not show differential engagement of the FAO of N-glycosylation pathways when compared to each other, while each of them shows strongly upregulated lipid metabolism and N-glycosylation in comparison to basal cells (Supplementary Figure 2d). These results suggest that FAO, as well as N-glycosylation, are upregulated during the differentiation of basal cells towards either of the differentiated cell fates.

Figure 2. differentiated cells disappear in presence of Etomoxir (2b), but a large portion of basal cells shifted to SSC-hi, suggesting that this drug promotes intracellular vesicles. What happens to basal cells in presence of Etomoxir?

To test if increased sideward scatter could be due to increased neutral lipid droplets, we stained with BODIPY 493/503 and found increased staining. These results are reported in Fig. 3c and in the text line 210-216.

In addition to q-PCR data, IF staining was also required to confirm for each lineage differentiation, including Muc5Ac and CCSP (2c). IF images must be shown in 2d. In 2e, human BEC cells are bronchial epithelial cells, not relevant to tracheal basal cells which is not helpful.

While we are convinced that PCR is the most sensitive and most quantifiable assay to determine cell presence in the cultures, we have added a new IF figure panel (Supplementary Figure 3a) which shows that etomoxir treatment prevented the appearance not only of MCCs but also of goblet cells (Muc5ac⁺) in mTEC cultures. Both micrographs and quantification are included in this figure. While CCSP IF staining works very well on trachea and lung sections (and is included in Supplementary Fig. 8), the quality of staining of mTEC or hBEC cultures is never satisfactory, with hazy stains and high background. We therefore have not included this.

Moreover, we now show in new Supplementary Figure 4 captions of the immunofluorescent staining against acetylated α -tubulin, as quantified in Figure 2d. Finally, as we outline above, current literature showed that human bronchial airways share similarities with murine tracheae in what concerns the pseudostratification of the epithelium and the cellular composition (reviewed by Hogan and colleagues (2010) and in Zepp and Morrissey (2019)). We thus believe that the use of hBEC cultures is justified and confers pre-clinical relevance to our study.

Figure 3, IF staining of CCSP and mucins are also required (3c).

Following the Reviewer's suggestion, we now include as Supplementary Figure 5a the staining of Muc5ac performed on mTEC cultures of *Cpt1a*-sufficient and *Cpt1a*-deficient mice. These data show a clear reduction in Muc5ac⁺ cells as quantified in this figure (Results line 242). As already explained above, CCSP stainings on mTEC do not allow for unequivocal identification of positive cells and is therefore not included here.

And nuclei staining must be also provided in ACT staining. Naphthalene is known to kill club cells, and ozone to kill ciliated cells. Naphthalene-induced or ozone-induced lung injury models should be used for *Cpt1a* KO mice to confirm in vitro findings on the regulatory role of *Cpt1a* on proximal airways.

Following the suggestion of Reviewer 1, we now included and quantified nuclear staining with DAPI across the new version of the manuscript (e.g. Supplementary figures 3, 4 and 5). Concerning the naphthalene approach suggested by Reviewer 1, we are sceptical about the technical viability of this strategy to address the reappearance of Ccsp⁺ cells. Naphthalene preferentially kills club cells due to their heightened expression of cytochrome P450 enzymes that can process naphthalene into a cytotoxic compound. However, the penetrance of this phenomenon is not complete, and therefore Ccsp-negative cells as well as some Ccsp⁺ cells will survive the treatment, thus hindering our efforts to quantify emergence of newly-developed Ccsp⁺ or indeed multiciliated cells. In contrast to naphthalene, polidocanol causes homogeneous injury of the airway epithelium, as shown by near-complete denudation of the tracheae in Supplementary figure 7, and thus we think it represents a superior model to test our hypothesis. Finally, the regulatory environment in the UK does not easily allow us to use the naphthalene and ozone models, and we do not currently have approval to use these.

As Reviewer 1 hints to use the alternative strategies of naphthalene and ozone to address the regulatory role of *Cpt1a* in the regeneration of proximal airways, we would like to reiterate that the protocol employed to induce epithelial injury, using polidocanol in 20uL of volume i.n., strictly limits epithelial damage to the proximal airways. The evidence for this is provided in the figure below, showing no measurable damage in the lung tissue, including large distal airways, which is in contrast to the denudation found in polidocanol-treated tracheae (see

below). Blinded scoring of the corresponding lung sections by a pathologist yielded consistently a score of zero for both control and polidocanol-treated lungs and distal airways. All in all, we trust that the additional *in vivo* data we provided here, as well as clarifications provided in this point-by-point reply, will answer the questions raised by Reviewer 1 regarding the important role of *Cpt1a* in proximal airways.

Polidocanol-induced injury is limited to the tracheal epithelium and does not extent to the lung of treated animals
(a) Haematoxylin & Eosin (H&E) stainings of tracheae and lungs from mock and polidocanol-treated *C57Bl/6J* mice at 2 days post treatment.

Figure 4. influenza infection has been demonstrated to induce dedifferentiation of club cells into basal cells (PMID:24196716). Did authors have a similar finding?

Reviewer 1 raises another interesting point. As shown in the paper cited here by Reviewer 1, for significant amounts of dedifferentiation of club cells into basal cells to happen, complete ablation of the basal cell pool is required, clearly indicating that in the presence of basal cells, these predominate in the repair process. In line with this, another study (Rawlins et al cell Stem Cell 2009) shows that only a very small proportion (0.34%) of Club cells dedifferentiate to express basal cell markers. Considering that our influenza models is well recognized to resemble physiological settings, we think that our data most likely addresses the effect of *Cpt1a* deletion on the ability of basal cells to differentiate, rather than a very rare event of dedifferentiation.

And influenza infection-induced pathologies restrict to distal airways and alveoli. If authors are interested in regulation of alveolar AT2 cell differentiation by FAO, organoid cultures with FAO inhibitors must be conducted.

As we previously stated, our focus here is not the role of FAO on alveolar cell regeneration, rather in the role of FAO in regulating airway epithelial regeneration. This is of high clinical relevance, as an extensive body of literature showed that influenza virus induces severe

pathology in proximal and distal airways (the decades-long literature on this is reviewed in Morens and Taubenberger 2008 and Flerlage, Shultz-Cherry 2021). Our own data also demonstrated cellular pathology driven by influenza infection in the proximal airway epithelium (see TUNEL staining in Davidson et al Nature Comm.s 2014). Finally, using the flow cytometry staining described in Supplementary Figure 6, we could separate the analysis of small airway epithelial cells from the ones located in the alveolar compartment, ultimately allowing us to describe the role of FAO in the regeneration of airway epithelium upon influenza virus infection, a process of high clinical relevance given the extensive literature on influenza-induced airway pathology. To re-iterate, we do not study alveolar or AT2 regeneration in this paper.

There are several epithelial progenitor cells in small airways including variant club cells, bronchioalveolar stem cells, etc. what do SAECs stand for in this aspect?

What are differentiated cells then do authors refer to here? IF staining of in vitro cultures of in vivo experiments must be conducted to verify.

As introduced in the text, SAEC stands for small airway epithelial cells, and both nomenclature and staining strategy follow (Quantius, J. et al. Influenza Virus Infects Epithelial Stem/Progenitor Cells of the Distal Lung: Impact on Fgfr2b-Driven Epithelial Repair. PLoS Pathog 12, e1005544, doi:10.1371/journal.ppat.1005544), again as was already cited in the text. In figure 1c of this citation, CD24 high cells are shown to be b-tubulin or Muc5ac or CC10 positive, conclusively showing that this gating strategy identifies club cells, multiciliated and goblet cells. As our staining strategy is identical, we do not see the need to further verify the identity of these cells as demonstrated in a highly-cited paper that we quote here.

Extended figure 4, please also show tracheal and alveolar epithelia after polidocanol. If this injury model is not specific to certain epithelial cells in the lung, it will be difficult to evaluate functions of their mother cells?

As stated above and supported by lung and trachea sections, our protocol of instillation of polidocanol is optimized to only cause injury in the airways. However, following the comment of Reviewer 1, we included representative images of lung and trachea upon polidocanol instillation. Tracheae show loss of the columnar epithelium two days after polidocanol instillation (see arrowheads). On the contrary, lungs of polidocanol-instilled mice do not show any obvious sign of ongoing inflammation as compared to control mice (see figure above included in this point-by-point reply). We have also included in Supplementary figure 8 a staining for CCSP, to show conclusively that in the absence of Cpt1a, neither multiciliated nor club cells redevelop (text line 337-339).

To conclude, we consciously chose a model to cause substantial and reproducible damage limited to the trachea, to be sure that the differentiation process would happen *de novo*, as opposed to residual semi-differentiated or differentiated cells confounding our findings. All experimental approaches we use in vitro and in vivo converge on a simple finding, that is that differentiation of cells from precursors in airway epithelia cannot be achieved in absence of FAO.

If alveolar part was included in the study, “airway” in the title of this manuscript seems to be inappropriate.

As previously stated, the alveoli are not the subject of our study. The title exactly reflects the content of the paper.

Reviewer #2 (Remarks to the Author):

In this manuscript, Dr. Crotta and team report their findings on the importance of a metabolic shift from glycolysis to fatty acid oxidation (FAO) during repair from airway epithelial cell injury. Specifically, they find that FAO is necessary for maintaining mitochondrial respiration while allowing glucose to be preferentially used in the hexosamine biosynthesis pathway, supporting protein glycosylation and epithelial cell differentiation during repair.

Overall, I found this to be a very well written manuscript reporting a detailed series of experiments that complement each other to form a comprehensive picture of the role of FAO in this pathology. The results appear to be consistent across several different models (robust to both the injury mechanism and the method of FAO modulation). The conclusions are highly consistent with the data presented. In addition, the area of research is important, as injury prevention is often not possible by the time of clinical presentation for many acute lung pathologies, so mechanisms to improve healing/repair are an important contribution.

Thanks for your enthusiasm for our study!

Major Comments:

My only major comment concerns the discussion. This section reads more like a summary of the results rather than a broader discussion of study relevance, importance, and comparison to existing literature on airway repair and metabolic response to lung injury. This strategy feels redundant, especially given the excellent summary statements presented in the results section, which keep the reader well caught-up with the pertinent findings even before reaching the discussion section. It seems that at most 1-2 paragraphs of the discussion should be needed to summarize the key findings, allowing the rest of the discussion to be converted to a broader conversation about how these findings fit with what is known and the potential novel implications for advancing the field of airway repair after acute lung injury.

We thank Reviewer 2 for the comments regarding our manuscript. We completely agree with Reviewer 2 in regard to the structure of the discussion. We have now reduced the extent of the parts dedicated to the summary of our findings, and expanded the ones to place our findings in the context of previous literature regarding the field of airway epithelial repair. We have also extended the discussion of metabolic changes in alveolar epithelia to contrast our findings.

Minor Comments:

1) Can the authors provide additional information on the mass spectrometry portion of the metabolic assay? Sample prep and data processing are well covered but I could not find information on the assay itself.

Following this comment of Reviewer 2, we now have improved the level of detail of our mass spectrometry protocols (line 683-696).

2) Can the authors discuss their choice to perform untargeted metabolomics rather than targeted (or even better fully quantitative) metabolite measurements? Particularly if the focus was on FAO there should be a limited number of metabolites that could be measured with high validity.

Reviewer 2 raises a valid point. We decided to perform untargeted metabolomics to get a broader picture of the metabolic profiles characterizing basal and differentiated cells.

For instance, we had no previous assumptions in regard to the kind of phospholipids that may be affected by the switch from glycolysis to FAO. Since all the phospholipids contain roughly

the same fatty acids that can equally well serve as substrates for FAO, we decided for the untargeted approach to obtain the most comprehensive picture of the phospholipid landscapes of airway epithelial cells. Similarly, the untargeted approach allowed us to collect broad information regarding the polar metabolites possibly affected by FAO engagement. It turned out that metabolites directly linked to FAO, such as acetyl-CoA and citrate, did not show remarkable changes. On the other hand, nucleobases and nucleosides, normally not included in a FAO-targeted approach, were instead substantially affected during the progression from basal cells to differentiated cells.

3) I could not find a statistical analysis section that covered the metabolomic analyses (or really that covered any of the data presented other than the RNA sequencing). Further details would be helpful.

Following this remark, we went through our original manuscript. Statistical details have been added in the respective figure legends, for ease of interpretation. We now made sure that this is clearly stated in the extended section dedicated to the statistical analysis in Methods (line 729-735).

Reviewer #3 (Remarks to the Author):

This is a comprehensive, rigorous manuscript describing a heretofore not well-defined impact of fatty acid oxidation (FAO) on airway epithelial cell fate in mice. The approach of the authors was to compare the metabolic profiles of airway stem cells (mostly basal cells) with their differentiated progeny. They discovered clear differences in the lipid profiles and then undertook numerous mechanistic studies both in vitro and in vivo that came to the conclusion that FAO is required for differentiation of progenitors to differentiated airway cells, especially ciliated cells. I find the conclusion and supporting data convincing and my comments mostly minor. The limitation of this study is mainly that the authors had to go to extremes to uncover a phenotype and a convincing roll for FAO in airway cell differentiation. The roll of FAO emerged only after they completely blocked import of palmitoyl-CoA into mitochondria by deleting or pharmacologically blocking the rate limiting mitochondrial transporter of palmitoyl-CoA (*cpt1a*). One wonders whether this scenario would ever occur in humans without congenital enzyme deficiency.

This is a very interesting point raised by Reviewer 3. We agree that genetic deletion of *Cpt1a* is a rather extreme scenario, when compared to clinical settings where *Cpt1a* expression may only be slightly affected. However, we would like to point out first that the inducible KO systems used here show incomplete penetrance, as indicated e.g. in figure 4b, where *Cpt1a* mRNA is reduced 4-5 times. In addition, *Cpt1a* deletion only prevents medium/long chain fatty acids from entering FAO. Despite the possibility for medium/short chain fatty acid to fuel FAO, we observed a remarkable phenotype, suggesting that incomplete perturbation of FAO may impair epithelial regeneration in the airways.

Nonetheless the data set are high quality, new to the field, and provide a clear take home message.

Specific criticisms:

1. The argument that the lung is low in glucose is mis-stated in the introduction. The provided reference refers only to the thin airway surface liquid layer (ASL). This should be stated specifically because otherwise there is no evidence or reason to believe glucose levels are low in the lung. It is true that most of the cells in the lung are mainly using respiration and not glycolysis for energy but this appears to be due mostly to the relatively high oxygen levels and not low glucose.

This is an excellent point raised by Reviewer 3. We now amended our introduction (line 54-55).

2. Fig 2e. The data show dramatic decrease in alpha-tubulin/ciliated cells but there are not controls to assure little toxicity with these treated human BEC cultures. The number of basal cells in mock and treated conditions should also be shown. In line with other cells studied here, one would expect more progenitors (mainly basal cells) in the treated cultures. This is a relevant point because in other settings blockade of FAO can lead to apoptosis in the lung (Yao et al, Am J Respir Cell Mol Biol 60:667, 2019.)

This is a valid point raised by Reviewer 3. Based on this comment, and the remark of Reviewer 1, we have added representative images of DAPI staining, and its relative quantification, of hBEC cultures upon etomoxir treatment in Supplementary Figure 4c, and we have added DAPI

stainings and/or quantifications in additional Supplementary figures (3a, 4a and b and 5a). We consistently find that genetic or pharmacological FAO blockade did not reduce total cell numbers in mTEC or hBEC cultures and induced an increase in basal cell number (Supplementary Fig.4c)

3. Extended Fig 4D. The *cpt1a* immunofluorescence is barely visible in the *cpt1a* ^{-/-} images of Region B even though there is acetylated tubulin in some cells. One cannot tell whether the cells expressing acetylated tubulin are the cells without detectable *cpt1a* (^{-/-}), as in other data, or if there is haploinsufficiency in some cells from partial recombination and that is sufficient to promote differentiation?

Following the suggestion of Reviewer 3, we re-present in new Supplementary Figure 8, staining data from the experiments shown in the original version of Supplementary Figure 4d. Staining of two consecutive slices of *Cpt1a*-deficient trachea, upon polidocanol treatment, with Abs against 1. Acetylated α -tubulin/Ccsp and 2. Acetylated α -tubulin/*Cpt1a*, clearly showed that areas that lacked *Cpt1a* failed to display expression of either acetylated α -tubulin or Ccsp. These data suggest, in the perfectly controlled setting of incomplete penetrance of the tamoxifen-driven *Cpt1a* deletion, that FAO is required to sustain differentiation of both multiciliated cells and secretory cells in the airway epithelium.

The uncertainty raised by this point gets to the relevance of all the data as it translates to normal or diseased lung function in humans. It seems unlikely that complete loss of *cpt1a* function ever occurs in humans without congenital deficiency.

We refer Reviewer 3 to our previous answer to the first comment, regarding the extent of the deletion of *Cpt1a* on the overall function of FAO.

We also would like to point out that a relatively small reduction of FAO, perhaps due to infections, pollutants or genetic polymorphisms, may build upon concurrent illnesses or individual predisposition in enhancing individual susceptibility to pulmonary diseases such as influenza virus infection.

However, hypoxia does frequently occur and would attenuate FAO. Is this degree of FAO suppression sufficient to affect airway basal cell to ciliated cell differentiation, perhaps partially?

Reviewer 3 raises an excellent point here. We performed gene expression analysis of selected genes involved in the differentiation of airway epithelial cells, as well as of selected genes involved in glycolytic and fatty acid metabolism. We found that exposure of mTEC cultures to hypoxia reduced the expression of genes linked to the differentiation of multiciliated cells (*Mcidas*), goblet (*Muc5ac*, *Muc5b*) and secretory cells (*Scgb1a1*). On the opposite, *Krt5*, marker of basal cells, showed elevated expression upon hypoxic conditions, as compared to mTEC cultures exposed to normoxia (see Figure below, panel a). Moreover, hypoxia impaired expression of genes involved in FAO (*Acad10*, *Acadm*, *Acaa2*, *Cpt1a* and *CD36*), while enhancing the expression of genes encoding for glycolytic enzymes (*Hk2*, *Ldha*). We also found that hypoxia prevented differentiation of multiciliated cells, as identified by staining of acetylated α -tubulin (Figure below, panels b and c).

Hypoxia reduces expression of genes related to FAO and blocks epithelial cell differentiation

(a) qPCR analysis of mRNA expression levels of the indicated genes in mTEC cultures grown under normoxic or hypoxic (0.5% O₂) conditions, at ALI day 7. Values are normalized to *Hprt1* expression and shown as fold change relative to normoxic cultures. Data show mean \pm SD and are representative of three independent experiments. Statistics were performed using multiple *t* tests. (b) Fluorescence microscopy analysis and quantification of acetylated α -tubulin in normoxic or hypoxic cultures, analyzed at ALI day 10. Cells were analyzed at ALI day 10. Data show mean \pm SD and are representative of three independent experiments. Statistics were performed using two-tailed unpaired Student's *t* test. Fluorescence intensity of mock-treated cultures was set as 100%.

Reviewer #1 (Remarks to the Author):

No more comments.

Reviewer #3 (Remarks to the Author):

The revised version of the manuscript by Crotta et al addresses many of the queries raised previously. The data showing a drastic reduction of fatty acid oxidation (FAO) attenuates tracheal airway basal cell differentiation, at least partly through attenuation of protein glycosylation rather than fall in ATP or toxicity, seems solid and the conclusions justified. My enthusiasm all along has been limited by the relevance of the drastic measures needed to block FAO in vitro and in vivo for the differentiation phenotype to human biology. However, in re-reading the manuscript I come away thinking the authors may have missed a mechanistic link that could explain much of the data and clarify at least one specific pathway that is being regulated by FAO. O-glycosylation is absolutely required for Notch activity and the authors nicely show protein glycosylation is attenuated in the setting of FAO blockade. This raises the possibility that blockade of FAO is functioning by attenuating Notch signaling. The authors never mention Notch. Airway epithelial cell differentiation from basal cells is previously reported to be Notch dependent (Rock et al Cell Stem Cell, 2011) and similar findings have been reported by many other investigators, though depending on the degree of inhibition differentiation may be skewed toward either increase or loss of ciliated cells and consistently loss of secretory and goblet cells. I hesitate to ask for additional data but I think the authors would be well served to test whether the Notch intracellular domain (NICD) is partially or completely lost in the face of FAO blockade. Linking the findings to a specific morphogen known to regulate epithelial differentiation would increase the impact of this work.

Point-by-point reply to reviewers' requests

REVIEWERS' COMMENTS

Reviewer #1 (Remarks to the Author):

No more comments.

Our reply: Thanks

Reviewer #3 (Remarks to the Author):

The revised version of the manuscript by Crotta et al addresses many of the queries raised previously. The data showing a drastic reduction of fatty acid oxidation (FAO) attenuates tracheal airway basal cell differentiation, at least partly through attenuation of protein glycosylation rather than fall in ATP or toxicity, seems solid and the conclusions justified. My enthusiasm all along has been limited by the relevance of the drastic measures needed to block FAO in vitro and in vivo for the differentiation phenotype to human biology. However, in re-reading the manuscript I come away thinking the authors may have missed a mechanistic link that could explain much of the data and clarify at least one specific pathway that is being regulated by FAO. O-glycosylation is absolutely required for Notch activity and the authors nicely show protein glycosylation is attenuated in the setting of FAO blockade. This raises the possibility that blockade of FAO is functioning by attenuating Notch signaling. The authors never mention Notch. Airway epithelial cell differentiation from basal cells is previously reported to be Notch dependent (Rock et al Cell Stem Cell, 2011) and similar findings have been reported by many other investigators, though depending on the degree of inhibition differentiation may be skewed toward either increase or loss of ciliated cells and consistently loss of secretory and goblet cells. I hesitate to ask for additional data but I think the authors would be well served to test whether the Notch intracellular domain (NICD) is partially or completely lost in the face of FAO blockade. Linking the findings to a specific morphogen known to regulate epithelial differentiation would increase the impact of this work.

Our reply: We agree that nailing the mechanism that explains the block in development of secretory and multiciliated cells would be desirable, and we also agree that notch is an interesting candidate here. Glycosylation of notch is a complex and expanding field, and notch involvement differs at different developmental points. We therefore feel that, while interesting, additional experiments assessing notch glycosylation would be beyond this study. We have however included the possibility that notch glycosylation is part of the underlying mechanism in the discussion section (lines 521-530).